# Deep Learning for Protein-Ligand Docking: Are We There Yet?

## Abstract

The effects of ligand binding on protein structures and their *in vivo* functions carry numerous implications for modern biomedical research and biotechnology development efforts such as drug discovery. Although several deep learning (DL) methods and benchmarks designed for protein-ligand docking have recently been introduced, to date no prior works have systematically studied the behavior of docking methods within the *broadly applicable* context of (1) using predicted (apo) protein structures for docking (e.g., for applicability to unknown structures); (2) docking multiple ligands concurrently to a given target protein (e.g., for enzyme design); and (3) having no prior knowledge of binding pockets (e.g., for unknown pocket generalization). To enable a deeper understanding of docking methods' real-world utility, we introduce PoseBench, the first comprehensive benchmark for *broadly applicable* protein-ligand docking. PoseBench enables researchers to rigorously and systematically evaluate DL docking methods for apo-to-holo protein-ligand docking and protein-ligand structure generation using *both* single and multi-ligand benchmark datasets, the latter of which we introduce for the first time to the DL community. Empirically, using PoseBench, we find that (1) DL methods consistently outperform conventional docking algorithms; (2) most recent DL docking methods fail to generalize to multi-ligand protein targets; and (3) training DL methods with physics-informed loss functions on diverse clusters of protein-ligand complexes is a promising direction for future work. Code, data, tutorials, and benchmark results are available at `https://anonymous.4open.science/r/PoseBench-2CD8`.

## 1 Introduction

The field of drug discovery has long been challenged with a critical task: determining the structure of ligand molecules in complex with proteins and other key macromolecules (Warren et al., 2012). As accurately identifying such complex structures (in particular multi-ligand structures) can yield advanced insights into the binding dynamics and functional characteristics (and thereby, the medicinal potential) of numerous protein complexes *in vivo*, in recent years, significant resources have been spent developing new experimental and computational techniques for protein-ligand structure determination (Du et al., 2016). Over the last decade, machine learning (ML) methods for structure prediction have become indispensable components of modern structure determination at scale, with AlphaFold 2 for protein structure prediction being a hallmark example (Jumper et al., 2021).

As the field has gradually begun to investigate whether proteins in complex with other types of molecules can faithfully be modeled with ML (and particularly deep learning (DL)) techniques (Dhakal et al., 2022; Harris et al., 2023; Krishna et al., 2024), several new works in this direction have suggested the promising potential of such approaches to protein-ligand structure determination (Corso et al., 2022; Lu et al., 2024; Qiao et al., 2024; Abramson et al., 2024). Nonetheless, to date, it remains to be shown whether such DL methods can adequately generalize in the context of *apo* (i.e., unbound) protein structures and multiple interacting ligand molecules (e.g., which can alter the chemical functions of various enzymes) as well as whether such methods are more accurate than traditional techniques for protein-ligand structure determination (for brevity hereafter referred to interchangeably as structure generation or docking) such as template-based (Pang et al., 2023) or molecular docking software tools (Xu et al., 2023).

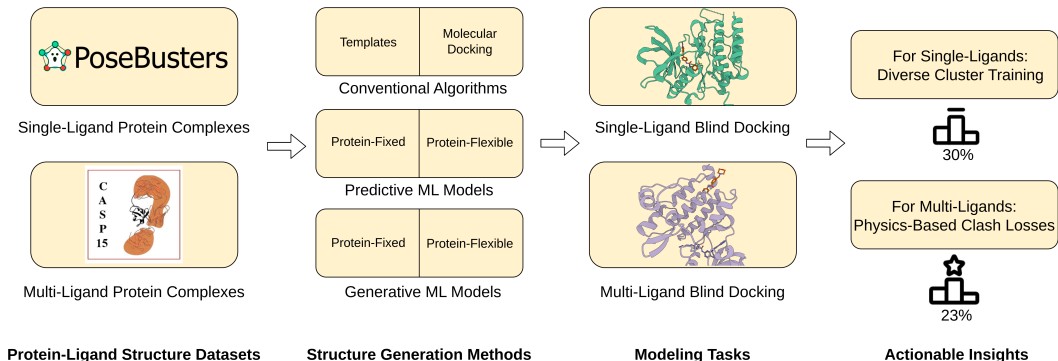

Figure 1: Overview of POSEBENCH, our comprehensive benchmark for *broadly applicable* ML modeling of single and multi-ligand protein complex structures in the context of apo (predicted) protein structures without known binding pockets (i.e., blind docking).

To bridge this knowledge gap, our contributions in this work are as follows:

- We introduce the first unified benchmark for protein-ligand structure generation that evaluates the performance of both recent DL-based methods as well as conventional methods for single and *multi*-ligand docking, which suggests that DL methods consistently outperform conventional docking techniques.

- In contrast to several recent works on protein-ligand docking (Buttenschoen et al., 2024; Corso et al., 2024a), the benchmark results we present in this work are all within the context of high accuracy *apo* (i.e., AlphaFold 3-predicted) protein structures without known binding pockets, which notably enhances the broad applicability of this study's findings.

- Our newly proposed benchmark, POSEBENCH, enables specific insights into necessary areas of future work for accurate and generalizable protein-ligand structure generation, including that physics-informed inter-ligand clash losses seems to be *key* to generalizing to multi-ligand docking targets.

- Our benchmark's results also highlight the importance of considering rigorous (e.g., structure-based) dataset splits when training future DL docking methods and measuring their ability to recapitulate the ground-truth distributions of protein-ligand interactions within benchmark datasets.

## 2 RELATED WORK

**Structure prediction of protein-ligand complexes.** The field of DL-driven protein-ligand structure determination was largely sparked with the development of geometric deep learning methods such as EquiBind (Stärk et al., 2022) and TANKBind (Lu et al., 2022) for direct (i.e., regression-based) prediction of bound ligand structures in protein complexes. Notably, these predictive methods could estimate localized ligand structures in complex with multiple protein chains as well as the associated complexes' binding affinities. However, in addition to their limited predictive accuracy, they have more recently been found to frequently produce steric clashes between protein and ligand atoms, notably hindering their widespread adoption in modern drug discovery pipelines.

**Protein-ligand structure generation and docking.** Shortly following the first wave of predictive methods for protein-ligand structure determination, DL methods such as DiffDock (Corso et al., 2022) demonstrated the utility of a new approach to this problem by reframing protein-ligand docking as a generative modeling task, whereby multiple ligand conformations can be generated for a particular protein target and rank-ordered using a predicted confidence score. This approach has inspired many follow-up works offering alternative formulations of this generative approach to the problem (Lu et al., 2024; Plainer et al., 2023; Zhu et al., 2024), with some of such follow-up works also being capable of accurately modeling protein flexibility upon ligand binding or predicting binding affinities to a high degree of accuracy.

**Benchmarking efforts for protein-ligand complexes.** In response to the large number of new methods that have been developed for protein-ligand structure generation, recent works have introduced several new datasets and metrics with which to evaluate newly developed methods, with some of such benchmarking efforts focusing on modeling single-ligand protein interactions (Buttenschoen et al., 2024; Durairaj et al., 2024) and with others specializing in the assessment of multi-ligand protein interactions (Robin et al., 2023). One of the primary aims of this work is to bridge this gap by systematically assessing a selection of the latest (pocket-blind) structure generation methods within both interaction regimes in the context of unbound protein structures and *ab initio* complex structure prediction, efforts we describe in greater detail in the following section.

## 3 POSEBENCH

The overall goal of POSEBENCH, our newly proposed benchmark for protein-ligand structure generation, is to provide the ML research community with a centralized resource with which one can systematically measure, in a variety of macromolecular contexts, the methodological advancements of new DL methods proposed for this problem. In the remaining sections, we describe POSEBENCH's design and composition (as illustrated in Figure 1), how we have used POSEBENCH to evaluate several recent DL methods (as well as conventional algorithms) for protein-ligand structure modeling, and what actionable insights we can derive from POSEBENCH's benchmark results with these latest DL methods.

### 3.1 PREPROCESSED DATASETS

POSEBENCH provides users with four datasets with which to evaluate existing or new protein-ligand structure generation methods, the Astex Diverse and PoseBusters Benchmark (DockGen) datasets previously curated by Buttenschoen et al. (2024) ((Corso et al., 2024a)) as well as the CASP15 protein-ligand interaction (PLI) dataset that we have manually curated in this work.

**Astex Diverse dataset.** The Astex Diverse dataset (Hartshorn et al., 2007) is a collection of 85 protein-ligand complexes composed of various drug-like molecules known to be of pharmaceutical or agrochemical interest, where a single representative ligand is present in each complex. This dataset can be considered an easy benchmarking set for many DL-based docking methods in that several of its proteins are known to overlap with the commonly used PDBBind (time-split) training dataset. Nonetheless, including this dataset for benchmarking allows one to determine the performance "upper bound" of each method's docking capabilities for single-ligand protein complexes.

To perform *apo* docking with this dataset, we used AlphaFold 3 (Abramson et al., 2024) to predict the complex structure of each of its proteins, where 5 of these 85 complexes were excluded from the effective benchmarking set due to being too large for docking with certain baseline methods on an 80GB NVIDIA A100 GPU. For the remaining 80 complexes, we then optimally aligned their predicted protein structures to the corresponding ground-truth (holo) protein-ligand structures using the PLI-weighted root mean square deviation (RMSD) alignment algorithm originally proposed by Corso et al. (2022).

**PoseBusters Benchmark dataset.** The PoseBusters Benchmark dataset (Buttenschoen et al., 2024) contains 308 recent protein-ligand complexes released from 2021 onwards. Like the Astex Diverse set, each complex in this dataset contains a single ligand for prediction. In contrast to Astex Diverse, this dataset can be considered a harder benchmark set since its proteins do not directly overlap with the commonly used PDBBind (time-split) training dataset composed of protein-ligand complexes with release dates up to 2019.

Likewise to Astex Diverse, for the PoseBusters Benchmark set, we used AlphaFold 3 to predict the *apo* complex structures of each of its proteins. After filtering out 28 complexes that certain baseline methods could not fit on an 80GB A100 GPU, we RMSD-aligned the remaining 280 predicted protein structures while optimally weighting each complex's protein-ligand interface in the alignment. For the **DockGen** dataset, we refer readers to Appendix H.1.

**CASP15 dataset.** To assess the multi-ligand modeling capabilities of recent methods for protein-ligand structure generation, in this work, we introduce a curated version of the CASP15 PLI dataset introduced as a first-of-its-kind prediction category in the 15th Critical Assessment of Structure

Table 1: POSEBENCH evaluation datasets for protein-(multi-)ligand structure generation.

| Name | Type | Source | Size (Total # Ligands) |
|---|---|---|---|
| Astex Diverse | Single-Ligand | (Hartshorn et al., 2007) | 80 |
| PoseBusters Benchmark | Single-Ligand | (Buttenschoen et al., 2024) | 280 |
| DockGen | Single-Ligand | (Corso et al., 2024a) | 91 |
| CASP15 | Multi-Ligand | | 102 (across 19 complexes) |
| | | | → 6 (13) single (multi)-ligand complexes |

Prediction (CASP) competition (Robin et al., 2023) held in 2022. The CASP15 PLI set is originally comprised of 23 protein-ligand complexes, where we subsequently filter out 4 complexes based on (1) whether the CASP organizers ultimately assessed predictions for the complexes; (2) whether they are nucleic acid-ligand complexes with no interacting protein chains; or (3) whether we could obtain a reasonably accurate prediction of the complex's multimeric protein chains using AlphaFold 3. Following this initial filtering step, we optimally align each remaining complex's predicted protein structures to the corresponding ground-truth protein-(multi-)ligand structures, weighting *each* of the complex's protein-ligand binding sites in the structural alignment.

The 19 remaining protein-ligand complexes, which contain a total of 102 (fragment) ligands, consist of a variety of ligand types including single-atom (metal) ions and large drug-sized molecules with up to 92 atoms in each (fragment) ligand. As such, this dataset is appropriate for assessing how well structure generation methods can model interactions between different (fragment) ligands in the same complex, which can yield insights into the (protein-ligand and ligand-ligand) steric clash rates of each method.

**Dataset similarity analysis.** For an investigation of the protein *sequence* similarity overlap between datasets such as the PoseBusters Benchmark set and the commonly-used PDBBind 2020 docking training dataset Liu et al. (2017), we refer interested readers to Buttenschoen et al. (2024). However, as a direct measure of the chemical and structural *pocket* similarity between PDBBind 2020 and the benchmark datasets employed in this work, in Appendix F.1, we analyze the different types and frequencies of protein-ligand pocket-level interactions natively found within the PDBBind 2020, Astex Diverse, PoseBusters Benchmark, DockGen, and CASP15 datasets, respectively, to quantify the diversity of the (predicted) interactions each dataset can be used to evaluate and to obtain an estimate of the (pocket-based) generalization challenges posed by each dataset. In short, we find that the DockGen and CASP15 benchmark datasets are the most dissimilar compared to PDBBind 2020.

## 3.2 FORMULATED TASKS

In this work, we have developed POSEBENCH to focus our analysis on the behavior of different DL methods for protein-ligand docking in a variety of macromolecular contexts (e.g., with or without inorganic cofactors present). With this goal in mind, below we formulate the structure generation tasks currently available in POSEBENCH.

**Single-ligand blind docking.** For single-ligand blind docking, each benchmark method is provided with a (multi-chain) protein sequence and an optional *apo* (predicted) protein structure as input along with a corresponding ligand SMILES string for each complex. In particular, no knowledge of the complex's protein-ligand binding pocket is provided to evaluate how well each method can (1) identify the correct binding pockets and (2) propose the correct ligand conformation within each predicted pocket.

**Multi-ligand blind docking.** For multi-ligand blind docking, each benchmark method is provided with a (multi-chain) protein sequence and an optional *apo* (predicted) protein structure as input along with the corresponding (fragment) ligand SMILES strings. As in single-ligand blind docking, no knowledge of the protein-ligand binding pocket is provided, which offers the opportunity to not only evaluate binding pocket and conformation prediction precision but also multimeric steric clash rates.

## 4 METHODS AND EXPERIMENTAL SETUP

**Overview.** Our benchmark is designed to explore answers to specific modeling questions for protein-ligand docking such as (1) which types of methods are best able to identify the correct binding pocket(s) in target proteins and (2) which types of methods most accurately produce multi-ligand structures without steric clashes? In the following sections, we describe in detail which types of methods we evaluate in our benchmark, what the input and output formats look like for each method, and how we evaluate each method's predictions for particular protein complex targets.

**Method categories.** As illustrated in Figure 1, we divide the benchmark methods included in POSEBENCH into one of three categories: (1) conventional algorithms, (2) predictive (i.e., regression-based) ML algorithms, and (3) generative (i.e., distributional) ML algorithms.

As representative algorithms for conventional protein-ligand docking, we include AutoDock Vina (v1.2.5) (Trott & Olson, 2010) as well as a template-based modeling method for ligand-protein complex structure prediction (TULIP) that we incorporate in this work to compare modern DL docking methods to the most common types of traditional docking algorithms (e.g., in the CASP15 competition (Xu et al., 2023)). For completeness, in Appendix G, we include a detailed description of the TULIP algorithm to provide interested readers with historical context regarding how such traditional docking techniques have typically been designed.

To represent predictive ML docking algorithms, we include FABind (Pei et al., 2024) as well as the recently released version of RoseTTAFold 2 for all-atom structural modeling (i.e., RoseTTAFold-All-Atom) (Krishna et al., 2024). Lastly, for generative ML docking algorithms, we include DynamicBind (Lu et al., 2024), NeuralPLexer (Qiao et al., 2024), Chai-1 (Chai-Discovery, 2024), and DiffDock-L (Corso et al., 2024a), the latest version of DiffDock, which is designed with pocket generalization as a key aim (n.b., through its use of ECOD (Cheng et al., 2014) structure-based cluster sampling). Notably, AlphaFold 3 (Abramson et al., 2024) does not currently support *generic* SMILES string inputs, so we cannot benchmark it.

Additionally, we provide a method ensembling baseline (Ensemble) that uses (multi-)ligand structural consensus ranking (Con) (Roy et al., 2023) to rank its ligand structure predictions selected from the (intrinsically method-ranked) top-3 ligand conformations produced by a subset of the DL baseline methods of this work (i.e., DiffDock-L, DynamicBind, NeuralPLexer, and RoseTTAFold-AA). This ensembling baseline is included to answer the question, "Which of these DL methods produces the most consistent conformations in interaction with a protein complex?".

**Input and output formats.**

1. Formats for conventional methods are as follows:

   a) Template-based (protein-fixed) methods such as **TULIP** are provided with an *apo* (predicted) protein structure and (fragment) ligand SMILES strings and are tasked with retrieving (PDB template (Bank, 1971)) ligand conformations residing in the same coordinate system as the given (predicted) protein structure following optimal molecular and structural alignment (Hu et al., 2018) with corresponding RDKit conformers of the input (query) ligand SMILES strings, where molecular similarity with the query ligands is used to rank-order the selected (PDB template) conformations.

   b) Molecular docking (protein-fixed) tools such as **AutoDock Vina**, which require specification of protein binding sites, are provided with not only a predicted protein structure but also the centroid coordinates of each (DiffDock-L-)predicted protein-ligand binding site residue. Such binding site residues are classified using a 4 Å protein-ligand heavy atom interaction threshold and using a 25 Å ligand-ligand heavy atom interaction threshold to define a "group" of ligands belonging to the same binding site and therefore residing in the same 25 $Å^3$-sized binding site input voxel for AutoDock Vina. For interested readers, for all four benchmark datasets, we also report results using P2Rank (Krivák & Hoksza, 2018) to predict AutoDock Vina's binding site centroid inputs.

2. Formats for predictive methods are as follows:

a) **FABind** is provided with a predicted protein structure as well as a ligand SMILES string, and it is then tasked with producing a (single) ligand conformation in complex with the given (fixed-structure) protein.

b) **RoseTTAFold-All-Atom (AA)** is provided with a (multi-chain) protein sequence as well as (fragment) ligand SMILES strings, and it is subsequently tasked with producing not only a (single) bound ligand conformation but also the bound (flexible) protein conformation (as a representative *ab initio* structure generation method).

3. Formats for generative methods are as follows:

a) **DiffDock-L** is provided with a predicted protein structure and (fragment) ligand SMILES strings and is then tasked with producing (multiple rank-ordered) ligand conformations (for each fragment) for the given (fixed-structure) protein. Note that DiffDock-L does not natively support multi-ligand SMILES string inputs, so in this work, we propose a modified inference procedure for DiffDock-L which *autoregressively* presents each (fragment) ligand SMILES string to the model while providing the same predicted protein structure to the model in each inference iteration (reporting for each complex the average confidence score over all iterations). Notably, as an inference-time modification, this sampling formulation permits multi-ligand sampling yet cannot model multi-ligand interactions directly and therefore often produces ligand-ligand steric clashes.

b) As a single-ligand generative (flexible) docking method, **DynamicBind** adopts the same input and output formats as DiffDock-L with the following exceptions: (1) the predicted input protein structure is now flexible in response to (fragment) ligand docking; (2) the autoregressive inference procedure we adapted from that of DiffDock-L now provides DynamicBind with its own most recently generated protein structure in each (fragment) ligand inference iteration, thereby providing the model with partial multi-ligand interaction context; and (3) iteration-averaged confidence scores *and* predicted affinities are reported for each complex. Nonetheless, for both DiffDock-L and DynamicBind, such modified inference procedures highlight the importance in future work of retraining such generative methods directly on multi-ligand complexes to address such inference-time compromises.

c) As a natively multi-ligand structure generation model trained using 3D molecular and protein data sources *and a physics-informed (Van der Waals) clash loss*, **NeuralPLexer** receives as its inputs a (multi-chain) protein sequence, a predicted protein (template) structure, as well as (fragment) ligand SMILES strings. The method is then tasked with producing multiple rank-ordered (flexible) protein-ligand structure conformations for each input complex, using the method's average predicted per-ligand heavy atom local Distance Difference Test (lDDT) score (Mariani et al., 2013) for rank-ordering.

d) Lastly, **Chai-1** serves as a multi-ligand structure generation model (akin to AlphaFold 3) trained on diverse sequence-based PDB clusters and AlphaFold 2-predicted structures along with AlphaFold 3-based training losses. Following its default settings for inference, the model receives as its inputs a (multi-chain) protein sequence and (fragment) ligand SMILES strings, with no template structures or multiple sequence alignments provided. The method is then tasked with producing multiple rank-ordered (flexible) protein-ligand structure conformations for each input complex, using the method's intrinsic ranking score (Abramson et al., 2024) for rank-ordering.

**Prediction and evaluation procedures.** Using the prediction formats above, the protein-ligand complex structures each method produces are subsequently evaluated using various structural accuracy and molecule validity metrics depending on whether the targets are single or multi-ligand complexes. We refer readers to Appendix D for formal definitions of POSEBENCH's structural metrics. Note that if a method's prediction raises any errors in subsequent scoring stages (e.g., due to missing entities or formatting violations), the prediction is excluded from the evaluation.

**Single-ligand evaluation.** For single-ligand targets, we report each method's percentage of (top-1) ligand conformations within 2 Å of the corresponding ground-truth ligand structure (RMSD $\leq 2$ Å) as well as the percentage of such "correct" ligand conformations that are also considered to be

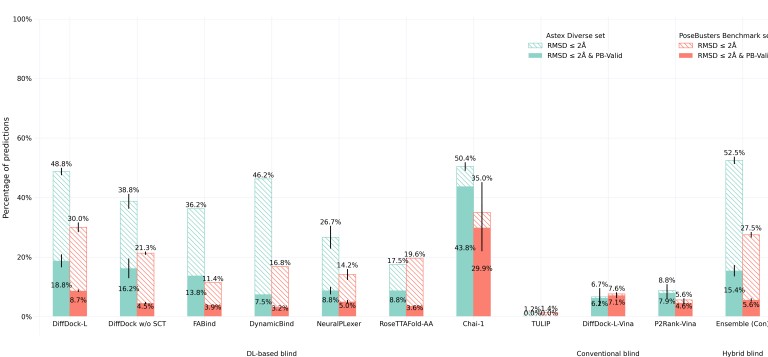

Figure 2: Astex & PoseBusters dataset results for successful single-ligand docking. RMSD $\leq$ 2 Å & PB-Valid denotes a method's percentage of ligand structures within 2 Å of the ground-truth ligand that also pass all PoseBusters filtering.

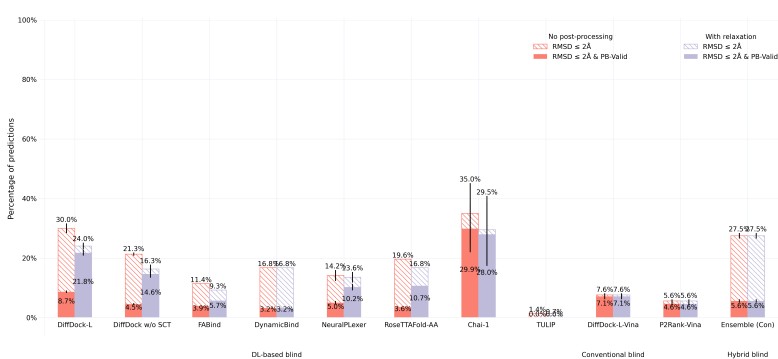

Figure 3: PoseBusters dataset results for successful single-ligand docking with relaxation. RMSD $\leq$ 2 Å & PB-Valid denotes a method's percentage of ligand structures within 2 Å of the ground-truth ligand that also pass all PoseBusters filtering.

chemically and structurally valid according to the PoseBusters software suite (Buttenschoen et al., 2024) (RMSD $\leq$ 2 Å & PB-Valid).

**Multi-ligand evaluation.** Following CASP15's official scoring procedure for protein-ligand complexes (Robin et al., 2023), for multi-ligand targets, we report each method's percentage of "correct" (binding site-superimposed) ligand conformations (RMSD $\leq$ 2 Å) as well as violin plots of the RMSD and PLI-specific lDDT scores of its protein-ligand conformations across all (fragment) ligands within the benchmark's multi-ligand complexes (see Appendix H for these plots). Notably, this final metric, referred to lDDT-PLI, allows one to evaluate specifically how well each method can model protein-ligand structural interfaces. In the remainder of this work, we will discuss our benchmark's results and their implications for the development of future structure generation methods.

## 5 RESULTS AND DISCUSSIONS

In this section, we present POSEBENCH's results for single and multi-ligand protein-ligand structure generation and discuss their implications for future work. Note that across all the experiments, for generative methods (or methods that use generative inputs to make their predictions), we report their performance metrics in terms of the mean and standard deviation across *three* independent runs of the method to gain insights into its inter-run stability and consistency. For interested readers, in Appendix C, we report the average runtime and memory usage of each baseline method to determine which methods are the most efficient for real-world docking applications.

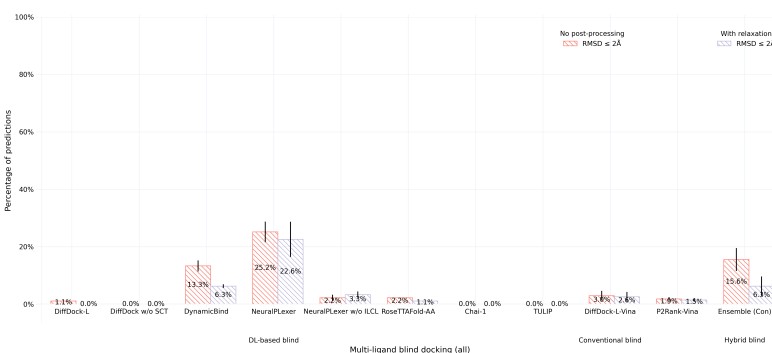

Figure 4: CASP15 dataset results for successful multi-ligand docking with relaxation. RMSD $\leq 2$ Å denotes a method's percentage of ligand structures within 2 Å of the ground-truth ligand.

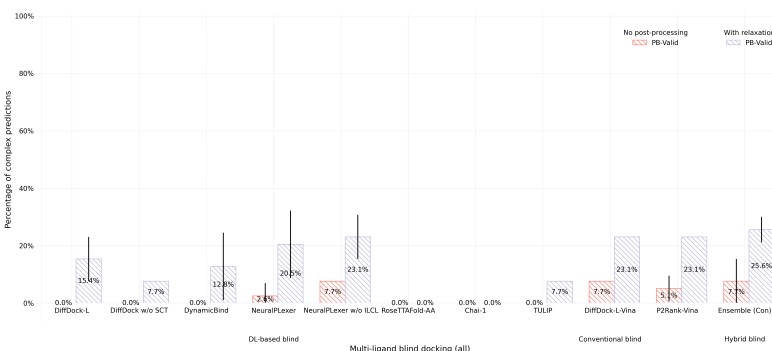

Figure 5: CASP15 dataset results for multi-ligand PoseBusters validity rates with relaxation. PB-Valid denotes a method's percentage of multi-ligand structures that pass all PoseBusters filtering.

### 5.1 TRAINING ON DIVERSE CLUSTERS SUPPORTS SINGLE-LIGAND DOCKING PERFORMANCE

We begin our investigations by evaluating the performance of each baseline method for single-ligand docking using the Astex Diverse and PoseBusters Benchmark datasets. Notably, for results on the PoseBusters Benchmark dataset (and subsequent datasets), we perform an additional analysis where we apply post-prediction (fixed-protein) relaxation to each method's generated ligand conformations using molecular dynamics simulations (Eastman & Pande, 2010), as originally proposed by Buttenschoen et al. (2024). Additionally, for interested readers, in Appendix H.1 we include DockGen benchmark results for flexible-protein relaxation as implemented by Lu et al. (2024).

As shown in Figures 2 and 3, Chai-1 and DiffDock-L (in particular, the version of DiffDock employing structural cluster training (SCT)) achieve the best overall performance across both of these *single-ligand* datasets in terms of its percentage of correct and valid generated ligand poses (i.e., RMSD $\leq 2$ Å & PB-Valid). To better understand this finding, in Appendix H.1, we find an even more striking instance where ablating SCT from DiffDock leads to considerably degraded docking performance for novel single-ligand protein targets. Furthermore, in the context of DockGen benchmarking, we find that Chai-1's performance closely matches the performance of DiffDock without SCT (both notably lower than that of DiffDock-L), suggesting that training on diverse *structural* clusters is particularly important for docking to *novel* protein pockets.

Following structural relaxation, closely behind in performance for the more challenging PoseBusters Benchmark dataset are the DL methods RoseTTAFold-AA and NeuralPLexer. Interestingly, without relaxation, AutoDock Vina combined with DiffDock-L's predicted binding pockets achieves the third-best performance on the PoseBusters Benchmark dataset, which suggests that (1) Chai-1 and DiffDock-L are currently the *only* single-ligand DL methods that present a better intrinsic understanding of biomolecular physics for docking than conventional modeling tools and (2) DiffDock-L is better at locating binding pockets than standard pocket predictors such as P2Rank. Overall, these

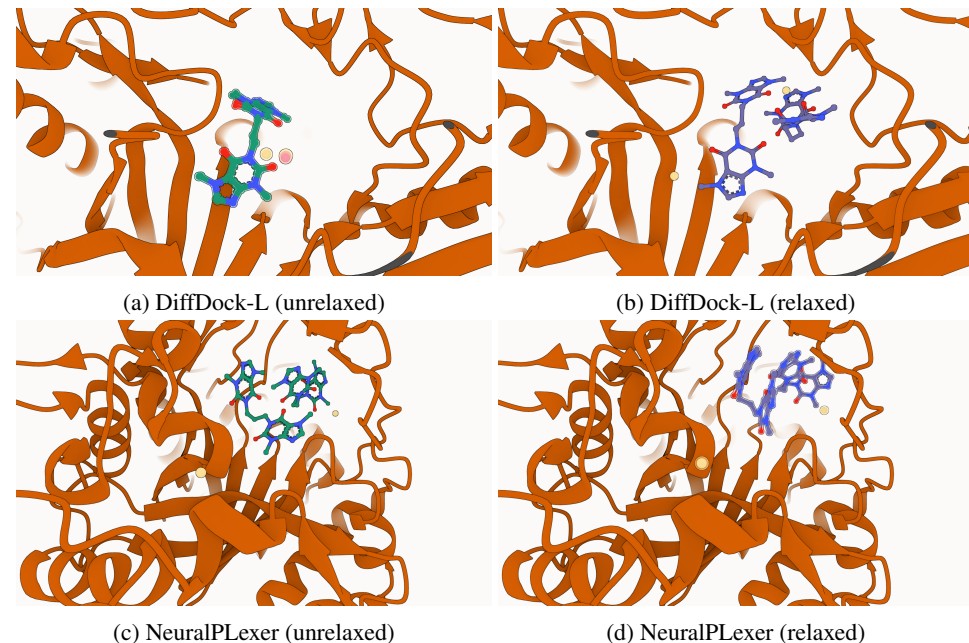

(a) DiffDock-L (unrelaxed)

(b) DiffDock-L (relaxed)

(c) NeuralPLexer (unrelaxed)

(d) NeuralPLexer (relaxed)

Figure 6: DiffDock-L and NeuralPLexer multi-ligand predictions for CASP15 target T1188.

results for the Astex Diverse and PoseBusters Benchmark datasets suggest that DL methods, combined with structural relaxation, outperform conventional methods for single-ligand docking and that training future DL methods using diverse sequence (and structure)-based clusters is a promising research direction for such docking tasks. For interested readers, in Appendix H.2, we report e.g., pocket-only PoseBusters Benchmark experiments and RMSD violin plots for both the Astex Diverse and PoseBusters Benchmark datasets, which suggest that Chai-1 and DiffDock-L primarily operate in sequence and structural representation spaces, respectively.

## 5.2 PHYSICS-INFORMED CLASH PENALIZATION IMPROVES MULTI-LIGAND DOCKING

We now turn to investigating the performance of various deep learning and conventional methods for *multi*-ligand docking. In contrast to the single-ligand docking results presented in Section 5.1, in Figure 4, we see a particular DL method, NeuralPLexer, stand out from all other methods in terms its multi-ligand docking performance. To better understand the factors contributing to its success, we also report results with a version of NeuralPLexer fine-tuned without its (original) van der Waals-based inter-ligand clash loss (ILCL) function (i.e., NeuralPLexer w/o ILCL), where these (ablation) results suggest that training NeuralPLexer with physics-based clash penalties has provided it with useful knowledge for successful multi-ligand docking. In contrast, all other baseline methods appear to produce only a handful of correctly docked multi-ligand poses. To more concretely understand why, in Appendix F.2, we plot the distribution of protein-ligand interactions produced by each baseline method for the CASP15 dataset, and we find that most methods struggle to correctly capture e.g., the distribution of hydrophobic interactions or Van der Waals contacts this dataset presents. Using CASP15 target T1188 as a case study, in Figure 6, we illustrate how this distributional mismatch often leads to methods such as DiffDock-L producing top-ranked predictions with multi-ligand steric clashes that must be (unoptimally) resolved using structural relaxation. To summarize, we find that these interaction-level distribution mismatches translate to poor multi-ligand docking performance for most baseline methods and that NeuralPLexer's inter-ligand clash loss has improved its ability to match the ground-truth distribution of CASP15 protein-ligand interactions for multi-ligand docking.

To further inspect each method's understanding of biomolecular physics for multi-ligand docking, in Figure 5 we report each method's percentage of predicted protein-ligand complexes (whether correct or not) for which all ligand conformations in the complex are jointly considered valid according to the PoseBusters software suite (i.e., PB-Valid). In short, in the context of multi-ligands, we find

that NeuralPLexer and AutoDock Vina are nearly tied in terms of their PoseBusters validity rates following structural relaxation and that Ensemble (Con) provides the best validity rates overall. To better understand this latter result, we note that NeuralPLexer's predictions seem to be among the most frequently selected by Ensemble (Con) for multi-ligand prediction targets (n.b., and conversely DiffDock-L for single-ligand targets), which suggests that NeuralPLexer consistently produces the highest percentage of valid ligand poses for a given multi-ligand complex, further supporting the notion that NeuralPLexer's multi-ligand training protocol has improved its understanding of protein-ligand binding patterns crucial for multi-ligand docking. For interested readers, in Appendix H.3, we report additional results e.g., in terms of lDDT-PLI and RMSD violin plots for both the total available CASP15 targets as well as those publicly available.

## CONCLUSIONS

In this work, we introduced POSEBENCH, the first deep learning (DL) benchmark for *broadly applicable* protein-ligand docking. Benchmark results with POSEBENCH currently suggest a *negative* answer to the question "Are we there yet (for structural drug discovery) with DL-based protein-ligand docking?". In this work, we have observed that while DL methods such as Chai-1 and DiffDock-L can identify the correct binding pockets in many single-ligand protein targets, most DL methods struggle to generalize to *multi*-ligand docking targets. Based on these results, for the development of future DL docking methods, we recommend researchers train new docking methods directly (1) on structurally clustered multi-ligand protein complexes available in new DL-ready biomolecular datasets (Abramson et al., 2024; Wang & Morehead, 2024) (2) using physics-informed inter-ligand steric clash penalties (Qiao et al., 2024). Key limitations of this study include its reliance on the accuracy of its predicted protein structures, its (currently) limited number of multi-ligand prediction targets available for benchmarking, and its inclusion of only a subset of all available protein-ligand docking baselines to focus on the most recent deep learning algorithms designed specifically for docking and structure generation. In future work, we aim to expand not only the number of baseline methods but also the number of available (CASP) multi-ligand targets while maintaining a diverse composition of heterogeneous (ionic) complexes. As a publicly available resource, POSEBENCH is flexible to accommodate new datasets and methods for protein-ligand structure generation.

**Availability.** The POSEBENCH codebase, documentation, and tutorial notebooks are available at https://anonymous.4open.science/r/PoseBench-2CD8 under a permissive MIT license, with further licensing discussed in Appendix A.

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

APPENDICES

## A  AVAILABILITY

The POSEBENCH codebase and tutorial notebooks are available under an MIT license at `https://anonymous.4open.science/r/PoseBench-2CD8`. Preprocessed datasets and benchmark method predictions are available on Zenodo under a CC-BY 4.0 license, of which the Astex Diverse and PoseBusters Benchmark datasets (Buttenschoen et al., 2024) are associated with a CC-BY 4.0 license; of which the DockGen dataset (Corso et al., 2024a) is available under an MIT license; and of which the CASP15 dataset (Robin et al., 2023), as a mixture of publicly and privately available resources, is partially licensed. In particular, 15 (4 single-ligand and 11 multi-ligand targets) of the 19 CASP15 protein-ligand complexes evaluated with POSEBENCH are publicly available, whereas the remaining 4 (2 single-ligand and 2 multi-ligand targets) are confidential and, for the purposes of future benchmarking and reproducibility, must be requested directly from the CASP organizers. Notably, the pre-holo-aligned protein structures predicted by AlphaFold 3 for these four benchmark datasets (available on Zenodo) must only be used in accordance with the Terms of Service provided by the AlphaFold Server. Lastly, our use of the PoseBusters software suite for molecule validity checking is permitted under a BSD-3-Clause license.

## B  BROADER IMPACTS

Our benchmark unifies protein-ligand structure generation datasets, methods, and tasks to enable enhanced insights into the real-world utility of such methods for accelerated drug discovery and energy research. We acknowledge the risk that, in the hands of "bad actors", such technologies may be used with harmful ends in mind. However, it is our hope that efforts in elucidating the performance of recent protein-ligand structure generation methods in various macromolecular contexts will disproportionately influence the positive societal outcomes of such research such as improved medicines and subsequent clinical outcomes as opposed to possible negative consequences such as the development of new bioweapons.

## C  COMPUTE RESOURCES

To produce the results presented in this work, we ran a high performance computing sweep that concurrently utilized 24 80GB NVIDIA A100 GPU nodes for 3 days in total to run inference with each baseline method three times (where applicable), where each baseline deep learning (DL) method required approximately 8 hours of GPU compute to complete its inference runs (except for FABind which completed its inference runs in the span of a couple hours). Notably, due to RoseTTAFold-All-Atom's significant storage requirements for running inference with its multiple sequence alignment databases, we utilized approximately 3 TB of solid-state storage space in total to benchmark all baseline methods. Lastly, in terms of CPU requirements, our experiments utilized approximately 64 concurrent CPU threads for AutoDock Vina inference (as an upper bound) and 60 GB of CPU RAM. Note that an additional 4-5 weeks of compute were spent performing initial (non-sweep) versions of each experiment during POSEBENCH's initial phase of development.

As a more formal investigation of the computational resources required to run each baseline method in this work, in Table 2 we list the average runtime (in seconds) and peak CPU (GPU) memory usage (in GB) consumed by each method when running them on a 25% subset of the Astex Diverse dataset.

## D  METRICS

In this work, we reference two key metrics in the field of structural bioinformatics: RMSD and lDDT. The RMSD between a predicted 3D conformation (with atomic positions $\hat{x}_i$ for each of the molecule's $n$ heavy atoms) and the ground-truth conformation ($x_i$) is defined as:

$$\text{RMSD} = \sqrt{\frac{1}{n}\sum_{i=1}^{n}\|\hat{x}_i - x_i\|^2}. \tag{1}$$

Table 2: The average runtime (in seconds) and peak memory usage (in GB) of each baseline method on a 25% subset of the Astex Diverse dataset (using an NVIDIA 80GB A100 GPU for benchmarking). The symbol - denotes a result that could not be estimated. Where applicable, an integer enclosed in parentheses indicates the number of samples drawn from a particular (generative) baseline method.

| Method | Runtime (s) | CPU Memory Usage (GB) | GPU Memory Usage (GB) |
|---|---|---|---|
| DiffDock-L (40) | 130.53 | 9.67 | 63.07 |
| FABind | 4.01 | 5.00 | 8.44 |
| DynamicBind (40) | 187.00 | 5.36 | 79.11 |
| NeuralPLexer (40) | 223.65 | 11.31 | 42.61 |
| RoseTTAFold-All-Atom | 862.60 | 49.78 | 78.97 |
| Chai-1 (5) | 297.77 | 37.49 | 73.90 |
| TULIP | - | - | - |
| DiffDock-L-Vina | 13.05 | 0.80 | 0.00 |
| P2Rank-Vina | 17.83 | 2.13 | 0.00 |
| Ensemble (Con) | - | - | - |

The lDDT score, which is commonly used to compare predicted and ground-truth protein 3D structures, is defined as:

$$\text{lDDT} = \frac{1}{N} \sum_{i=1}^{N} \frac{1}{4} \sum_{k=1}^{4} \left( \frac{1}{|\mathcal{N}_i|} \sum_{j \in \mathcal{N}_i} \Theta(|\hat{d}_{ij} - d_{ij}| < \Delta_k) \right), \tag{2}$$

where $N$ is the total number of heavy atoms in the ground-truth structure; $\mathcal{N}_i$ is the set of neighboring atoms of atom $i$ within the inclusion radius $R_o = 15$ Å in the ground-truth structure, excluding atoms from the same residue; $\hat{d}_{ij}$ ($d_{ij}$) is the distance between atoms $i$ and $j$ in the predicted (ground-truth) structure; $\Delta_k$ are the distance tolerance thresholds (i.e., 0.5 Å, 1 Å, 2 Å, and 4 Å); $\Theta(x)$ is a step function that equals 1 if x is true, and 0 otherwise; and $|\mathcal{N}_i|$ is the number of neighboring atoms for atom $i$.

As originally proposed by Robin et al. (2023), in this study, we adopt the PLI-specific variant of lDDT, which calculates lDDT scores to compare predicted and ground-truth protein-ligand complex structures following optimal structural alignment of the predicted and ground-truth protein-ligand binding pockets.

## E  DOCUMENTATION FOR DATASETS

Below, we provide detailed documentation for each dataset included in our benchmark, summarised in Table 1. Each dataset is freely available for download from the benchmark's accompanying Zenodo data record under a CC-BY 4.0 license. In lieu of being able to create associated metadata for each of our macromolecular datasets using an ML-focused library such as Croissant (Akhtar et al., 2024) (due to file type compatibility issues), instead, we report structured metadata for our preprocessed datasets using Zenodo's web user interface. Note that, for all datasets, we authors bear all responsibility in case of any violation of rights regarding the usage of such datasets.

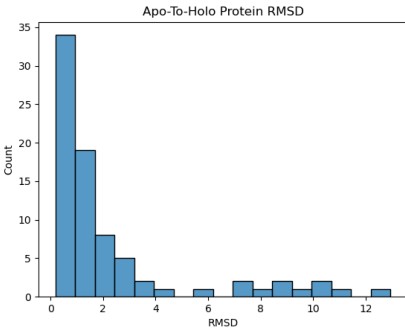

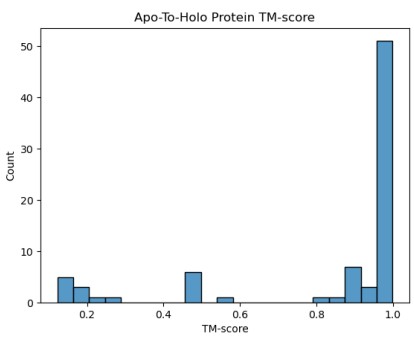

(a) RMSD of AlphaFold 3's predictions.

(b) TM-score of AlphaFold 3's predictions.

Figure 7: Accuracy of AlphaFold 3's predicted protein structures for the Astex Diverse dataset.

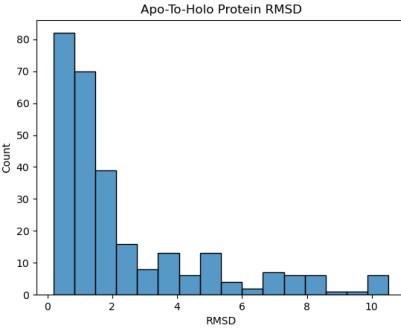

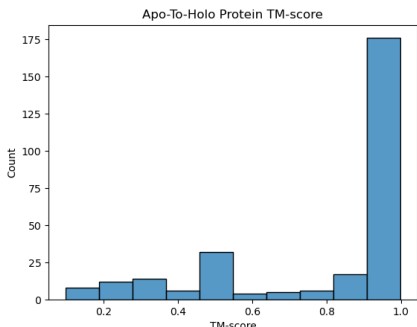

(a) RMSD of AlphaFold 3's predictions.

(b) TM-score of AlphaFold 3's predictions.

Figure 8: Accuracy of AlphaFold 3's predicted protein structures for the PoseBusters Benchmark dataset.

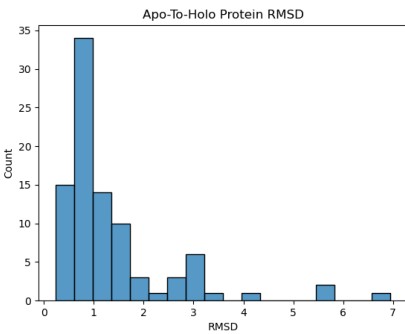

(a) RMSD of AlphaFold 3's predictions.

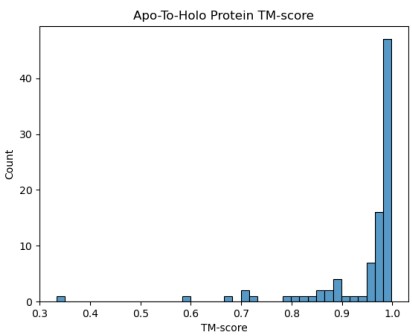

(b) TM-score of AlphaFold 3's predictions.

Figure 9: Accuracy of AlphaFold 3's predicted protein structures for the DockGen dataset.

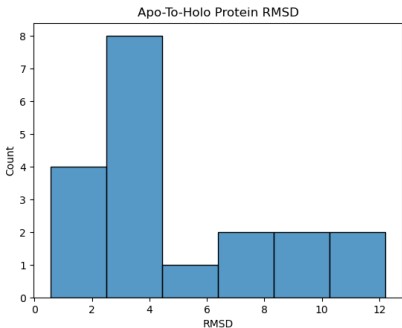

(a) RMSD of AlphaFold 3's predictions.

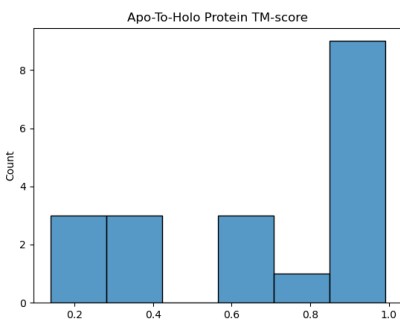

(b) TM-score of AlphaFold 3's predictions.

Figure 10: Accuracy of AlphaFold 3's predicted protein structures for the CASP15 dataset.

## E.1 ASTEX DIVERSE SET - SINGLE-LIGAND DOCKING
(DIFFICULTY: *Easy*)

A common drug discovery task is to screen several novel drug-like molecules against a target protein in rapid succession. The Astex Diverse dataset was originally developed with this application in mind, as it features many therapeutically relevant 3D molecules for computational modeling.

- **Motivation** Several downstream drug discovery efforts rely on having access to high-quality molecular data for docking.
- **Collection** For this dataset, which was originally compiled by Hartshorn et al. (2007), we adopt the version further prepared by Buttenschoen et al. (2024).
- **Composition** The dataset consists of 80 single-ligand protein complexes for which we could obtain high-accuracy predicted protein structures using AlphaFold 3. The accuracy of the AlphaFold 3-predicted structures is measured in terms of their RMSD and TM-score (Zhang & Skolnick, 2004) compared to the corresponding ground-truth (i.e., experimental) protein structures and is visualized in Figure 7. Notably, 79% of the predicted structures have an RMSD below 4 Å and a TM-score above 0.7, indicating that most of the dataset's proteins have a reasonably accurate predicted structure.
- **Hosting** Our preprocessed version of the dataset can be downloaded from the benchmark's Zenodo data record.
- **Licensing** We have released our preprocessed version of the dataset under a CC-BY 4.0 license. The original dataset is available under a CC-BY 4.0 license on Zenodo (Buttenschoen et al., 2023). The pre-holo-aligned protein structures predicted by AlphaFold 3 for this dataset (available on Zenodo) must only be used in accordance with the Terms of Service provided by the AlphaFold Server.
- **Maintenance** We will announce any errata discovered in or changes made to the dataset using the benchmark's GitHub repository at `https://anonymous.4open.science/r/PoseBench-2CD8`.
- **Uses** This dataset of holo (and predicted-apo) protein PDB and holo ligand SDF files can be used for single-ligand docking or protein-ligand structure generation.
- **Metric** Ligand RMSD $\leq$ 2 Å & PoseBusters-Valid (PB-Valid).

### E.2 POSEBUSTERS BENCHMARK SET - SINGLE-LIGAND DOCKING
(DIFFICULTY: *Intermediate*)

Like the Astex Diverse dataset, the PoseBusters Benchmark dataset was originally developed for docking individual ligands to target proteins. However, this dataset features a larger and more challenging collection of protein-ligand complexes for computational modeling.

- **Motivation** Data sources of challenging single-ligand protein complexes for molecular docking are critical for the development of future docking methods.

- **Collection** For this dataset, we adopt the version introduced by Buttenschoen et al. (2024).

- **Composition** The dataset consists of 280 single-ligand protein complexes for which we could obtain high-accuracy predicted protein structures using AlphaFold 3. The accuracy of the AlphaFold 3-predicted structures is measured in terms of their RMSD and TM-score (Zhang & Skolnick, 2004) compared to the corresponding ground-truth (i.e., experimental) protein structures and is visualized in Figure 8. Notably, 70% of the predicted structures have an RMSD below 4 Å and a TM-score above 0.7, indicating that most of the dataset's proteins have a reasonably accurate predicted structure.

- **Hosting** Our preprocessed version of the dataset can be downloaded from the benchmark's Zenodo data record.

- **Licensing** We have released our preprocessed version of the dataset under a CC-BY 4.0 license. The original dataset is available under a CC-BY 4.0 license on Zenodo (Buttenschoen et al., 2023). The pre-holo-aligned protein structures predicted by AlphaFold 3 for this dataset (available on Zenodo) must only be used in accordance with the Terms of Service provided by the AlphaFold Server.

- **Maintenance** We will announce any errata discovered in or changes made to the dataset using the benchmark's GitHub repository at `https://anonymous.4open.science/r/PoseBench-2CD8`.

- **Uses** This dataset of holo (and predicted-apo) protein PDB and holo ligand SDF files can be used for single-ligand docking or protein-ligand structure generation.

- **Metric** Ligand RMSD $\leq$ 2 Å & PoseBusters-Valid (PB-Valid).

### E.3 DOCKGEN SET - SINGLE-LIGAND DOCKING
(DIFFICULTY: *Challenging*)

The DockGen dataset was originally developed for docking individual ligands to target proteins in the context of novel protein binding pockets. As such, this dataset is useful for evaluating how well each baseline method can generalize to distinctly different binding pockets compared to those on which it commonly may have been trained.

- **Motivation** Data sources of protein-ligand complexes representing novel single-ligand binding pockets are critical for the development of generalizable docking methods.

- **Collection** For this dataset, we adopt the version introduced by Corso et al. (2024a).

- **Composition** The dataset originally consists of 189 single-ligand protein complexes, after which we perform additional filtering down to 91 complexes based on structure prediction accuracy ($< 5$ Å $C\alpha$ atom RMSD for the primary protein interaction chain). The accuracy of the AlphaFold 3-predicted structures is measured in terms of their RMSD and TM-score (Zhang & Skolnick, 2004) compared to the corresponding ground-truth (i.e., experimental) protein structures and is visualized in Figure 9. Notably, 95% of the predicted structures have an RMSD below 4 Å and a TM-score above 0.7, indicating the majority of the dataset's proteins have a reasonably accurate predicted structure.

- **Hosting** Our preprocessed version of the dataset can be downloaded from the benchmark's Zenodo data record.

- **Licensing** We have released our preprocessed version of the dataset under a CC-BY 4.0 license. The original dataset is available under an MIT license on Zenodo (Corso et al., 2024b). The pre-holo-aligned protein structures predicted by AlphaFold 3 for this dataset (available on Zenodo) must only be used in accordance with the Terms of Service provided by the AlphaFold Server.

- **Maintenance** We will announce any errata discovered in or changes made to the dataset using the benchmark's GitHub repository at `https://anonymous.4open.science/r/PoseBench-2CD8`.

- **Uses** This dataset of holo (and predicted-apo) protein PDB and holo ligand PDB files can be used for single-ligand docking or protein-ligand structure generation.

- **Metric** Ligand RMSD $\leq 2$ Å & PoseBusters-Valid (PB-Valid).

### E.4 CASP15 Set - Multi-Ligand Docking
(Difficulty: *Challenging*)

As the most complex of our benchmark's four test datasets, the CASP15 protein-ligand interaction dataset was created to represent the new protein-ligand modeling category in the 15th Critical Assessment of Structure Prediction (CASP) competition. Whereas the Astex Diverse and PoseBusters Benchmark datasets feature solely single-ligand protein complexes, the CASP15 dataset provides users with a variety of challenging organic (e.g., drug molecules) and inorganic (e..g., ion) cofactors for *multi*-ligand biomolecular modeling.

- **Motivation** Multi-ligand evaluation datasets for molecular docking provide the rare opportunity to assess how well baseline methods can model intricate protein-ligand interactions while avoiding troublesome protein-ligand and ligand-ligand steric clashes. Additionally, more accurate modeling of multi-ligand complexes in future works may lead to improved techniques for computational enzyme design and regulation (Stärk et al., 2023).

- **Collection** For this dataset, we manually collect each publicly and privately available CASP15 protein-bound ligand complex structure compatible with protein-ligand (e.g., non-nucleic acid) benchmarking.

- **Composition** The dataset consists of 102 (86) fragment ligands contained within 19 (15) separate (publicly available) protein complexes, of which 6 (2) and 13 (2) of these complexes are single and multi-ligand complexes, respectively. The accuracy of the dataset's AlphaFold 3-predicted structures is measured in terms of their RMSD and TM-score (Zhang & Skolnick, 2004) compared to the corresponding ground-truth (i.e., experimental) protein structures and is visualized in Figure 10. Notably, 42% of the predicted structures have an RMSD below 4 Å and a TM-score above 0.7, indicating a portion of the dataset's proteins have a reasonably accurate predicted structure. Given the much larger structural ensembles of this dataset's protein complexes compared to those of the other three benchmark datasets, we believe the accuracy of these predictions may be improved with advancements in machine learning modeling of biomolecular assemblies.

- **Hosting** Our preprocessed version of (the publicly available version of) this dataset can be downloaded from the benchmark's Zenodo data record.

- **Licensing** We have released our preprocessed version of the (public) dataset under a CC-BY 4.0 license. The original (public) dataset is free for download via the RCSB PDB (Bank, 1971). The pre-holo-aligned protein structures predicted by AlphaFold 3 for this dataset (available on Zenodo) must only be used in accordance with the Terms of Service provided by the AlphaFold Server.

- **Maintenance** We will announce any errata discovered in or changes made to the dataset using the benchmark's GitHub repository at `https://anonymous.4open.science/r/PoseBench-2CD8`.

- **Uses** This dataset of holo (and predicted-apo) protein PDB and holo ligand PDB files can be used for multi-ligand docking or protein-ligand structure generation.

- **Metric** (Fragment) Ligand RMSD $\leq$ 2 Å & (Complex) PoseBusters-Valid (PB-Valid).

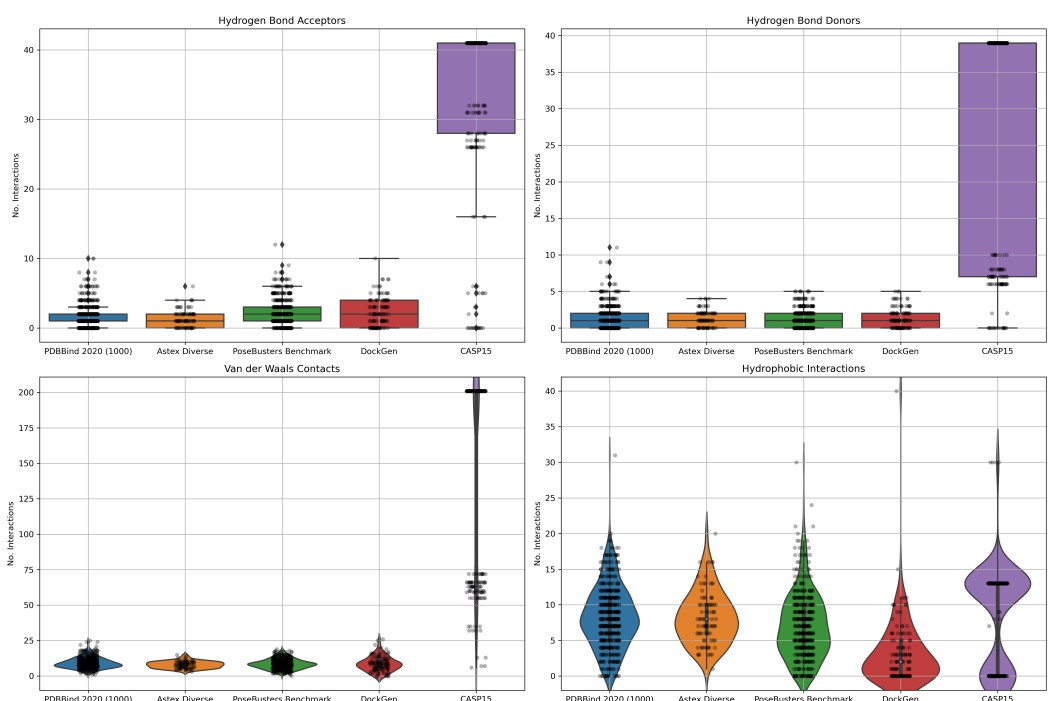

Figure 11: PDBBind 2020, Astex Diverse, PoseBusters Benchmark, DockGen, and CASP15 dataset comparative analysis of protein-ligand (pocket-level) interactions.

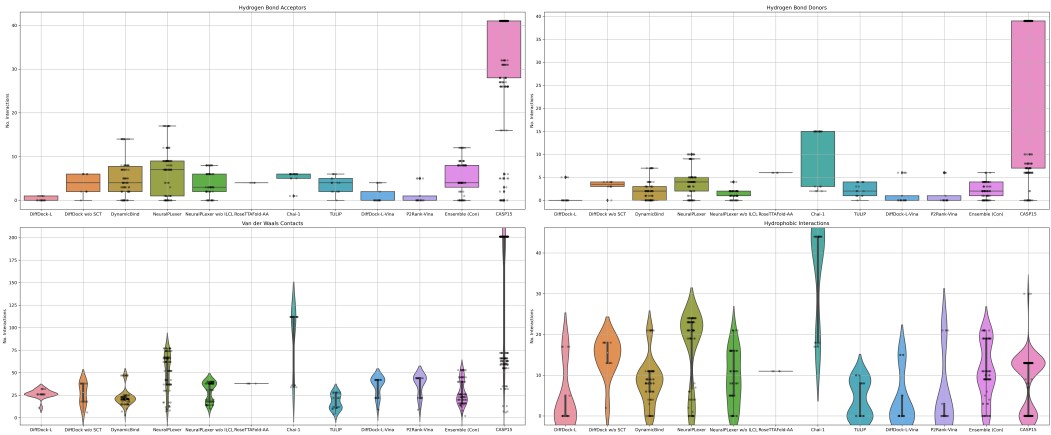

Figure 12: Comparative analysis of the protein-ligand (pocket-level) interactions within the CASP15 dataset and baseline method predictions.

# F ANALYSIS OF PROTEIN-LIGAND INTERACTIONS

## F.1 DATASET PROTEIN-LIGAND INTERACTION DISTRIBUTIONS

Inspired by a similar analysis presented in the PoseCheck benchmark (Harris et al., 2023), in this section, we study the frequency of different types of protein-ligand (pocket-level) interactions such as Van der Waals contacts and hydrophobic interactions occurring natively within (n.b., a size-1000 random subset of) the commonly-used PDBBind 2020 docking training set (i.e., PDBBind 2020 (1000)) as well as the Astex Diverse, PoseBusters Benchmark, DockGen, and CASP15 benchmark datasets, respectively. In particular, these measures allow us to better understand the diversity of interactions each baseline method within the POSEBENCH benchmark is tasked to model, within the context of each test dataset. Furthermore, these measures directly indicate which benchmark datasets are most *dissimilar* from commonly used training data for docking methods. Figure 11 displays the results of this analysis.

Overall, we find that the PDBBind 2020, Astex Diverse, and PoseBusters Benchmark datasets contain similar types and frequencies of interactions, with the PoseBusters Benchmark dataset containing slightly more hydrogen bond acceptors ($\sim$3 vs 1) and fewer Van der Waals contacts ($\sim$5 vs 8) on average compared to the PDBBind 2020 dataset. However, we observe a more notable difference in interaction types and frequencies between the DockGen and CASP15 datasets and the three other datasets. Specifically, we find these two benchmark datasets contain a notably different quantity of hydrogen bond acceptors and donors (n.b., $\sim$40 for CASP15), Van der Waals contact ($\sim$200 for CASP15), and hydrophobic interactions ($\sim$2 for DockGen) on average. As we will see in the DockGen benchmarking results reported in Appendix H.1, this latter observation supports our first key insight of this work, that training new docking methods on *structure-based dataset clusters* is a promising direction for future work on developing new pocket-generalizing docking methods.

Also particularly interesting to note is the CASP15 dataset's bimodal distribution of hydrophobic interactions, suggesting that the dataset contains two primary classes of interacting ligands giving rise to hydrophobic interactions. One possible explanation for this phenomenon is that the CASP targets, in contrast to the PDBBind, Astex Diverse, PoseBusters Benchmark, and DockGen targets, consist of a variety of both organic (e.g., drug-like molecules) and inorganic (e.g., metal) cofactors.

## F.2 PROTEIN-LIGAND INTERACTION DISTRIBUTIONS OF EACH BASELINE METHOD

Intrigued by the dataset interaction patterns in Figure 11, we further investigated the (predicted) protein-ligand interactions produced by each baseline method for the (multi-ligand) CASP15 dataset, to study which machine learning-based docking method can most faithfully reproduce the true distribution of protein-ligand interactions within this benchmark dataset. Our results in Figure 12 suggest, similar to our docking results in Figure 4, that NeuralPLexer demonstrates the best overall ability to recapitulate the complex interaction dynamics observed within this dataset, presenting the unique ability (among all baseline DL methods) to correctly capture the dataset's intricate (bimodal, top first-bottom second) interaction patterns within its hydrophobic interactions (Bogunia & Makowski, 2020; Sayyah et al., 2024). Combined with the CASP15 benchmarking results presented in Section 5 of the main text, this latter finding further supports our second key insight of this work, that the *physics-informed inter-molecular clash penalties* that DL methods such as NeuralPLexer employ have equipped them with physically relevant knowledge for multi-ligand docking.

# G ADDITIONAL METHOD DESCRIPTIONS

To better contextualize the benchmark's results comparing DL docking methods to conventional docking algorithms, in this section, we provide further details regarding how certain traditional docking methods in the benchmark leverage different sources of biomolecular data to predict protein-ligand interactions for given protein targets.

## G.1 TULIP

TULIP is a template-based modeling pipeline for predicting protein-ligand interactions that we present in the benchmark as a historical reference point to better contextualize the advances of

the latest DL methods for docking, as in the recent CASP15 competition template-based methods outperformed the DL docking methods that were available at the time (Xu et al., 2023). TULIP takes the target ligand's 3D initial conformer structure (Landrum et al., 2013), the predicted receptor protein structure, and identified template structures from MULTICOM (Liu et al., 2023) as inputs. TULIP first aligns the template structures containing ligands into the same geometric space as the predicted receptor structure using UCSF Chimera's matchmaker (Pettersen et al., 2004) in non-interactive mode. It then saves the superimposed template structures and their ligands relative to the predicted receptor structure in an output PDB file that is processed by PyRosetta's *is_ligand* function (Chaudhury et al., 2010) to identify template ligands by checking each residue against the Chemical Component Dictionary of the Protein Data Bank (PDB) (Westbrook et al., 2015). The extracted unique ligands from each template and the target ligand are converted into Morgan fingerprints (Zhou & Skolnick, 2024) to compute their Tanimoto molecular similarity (Bajusz et al., 2015) (n.b., a [0, 1] metric of increasing similarity). This step provides the initial binding location of the target ligand with respect to the receptor protein structure. Furthermore, to adjust the target ligand's binding pose and orientation by rotation and translation, TULIP uses LS-align (Hu et al., 2018) to align the target ligand with the template ligands of higher similarity through both flexible and rigid-body alignments. Between the flexible and rigid-body alignment outputs, TULIP selects the alignment with the lowest RMSD between the template and target ligands to obtain the predicted coordinates of the target ligand. Ligands with a distance greater than 6 Å from the protein surface are discarded. To handle multiple ligands with the same SMILES string, the identified ligands are grouped into n clusters, where n is the number of ligands with the same SMILES string. To compute the clusters, pairwise distances between the ligands are generated, and agglomerative clustering is used.

## H  ADDITIONAL RESULTS

In this section, we provide additional results for each baseline method using the Astex Diverse, PoseBusters Benchmark, and DockGen datasets as well as the CASP15 ligand targets. Note that for all violin plots listed in this section, we curate them using combined results across each method's three independent runs (where applicable), in contrast to this section's bar charts where we instead report mean and standard deviation values across each method's three independent runs.

### H.1  DOCKGEN RESULTS

**DockGen dataset.** The DockGen dataset (Corso et al., 2024a) contains 189 diverse single-ligand protein complexes, each representing a novel type of protein-ligand binding pocket. This dataset can be considered the most difficult single-ligand benchmark set since its protein binding sites are distinctly different from those commonly found in the training datasets of most deep learning-based docking methods to date.

For this dataset, we once again used AlphaFold 3 to predict the *apo* complex structures of each of its proteins. We performed additional filtering down to 91 of the dataset's complexes, as using AlphaFold 3 not all 189 of its protein complex structures could be accurately predicted (i.e., achieving $< 5$ Å C$\alpha$ atom RMSD for the primary protein interaction chains). After predicting each structure, we RMSD-aligned these *apo* structures while optimally weighting each complex's protein-ligand interface in the alignment.

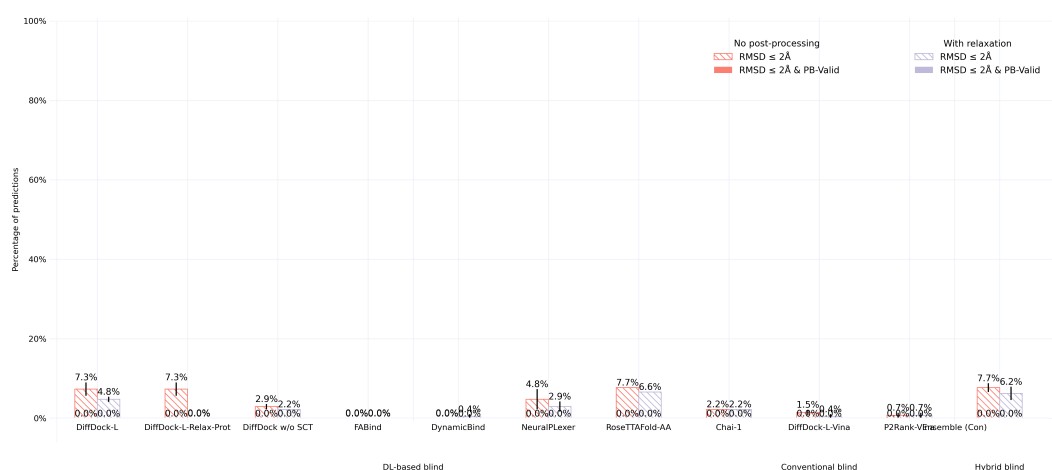

Figure 13: DockGen dataset results for successful single-ligand docking with relaxation.

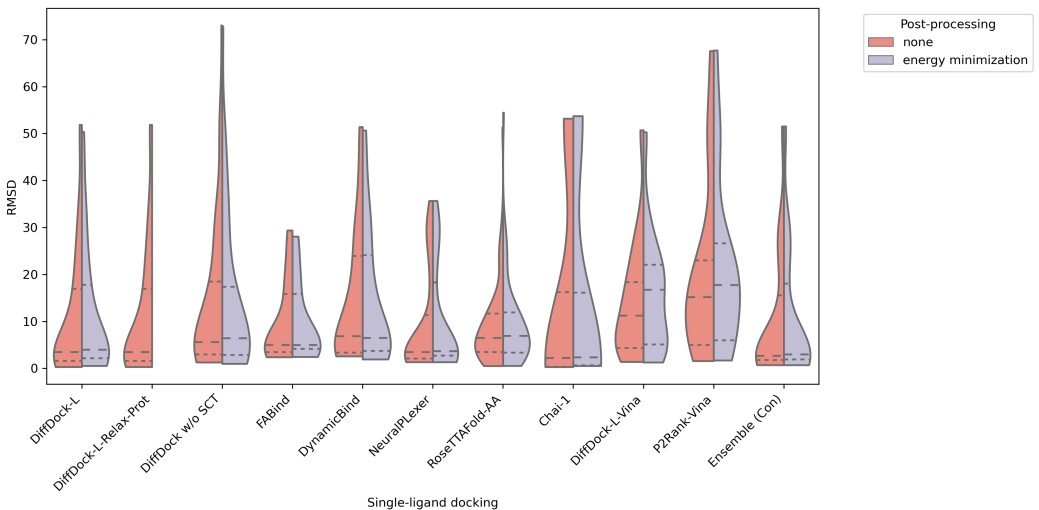

Figure 14: DockGen dataset results for single-ligand docking RMSD.

**Benchmark results.** Figures 13 and 14 reveal that DiffDock-L, RoseTTAFold-AA, and NeuralPLexer provide the best pocket generalization capabilities compared to all other baseline methods. Moreover, similar to the PoseBusters Benchmark dataset results in Section 5 of the main text, the results for DiffDock-L-Vina and P2Rank-Vina here further suggest that DiffDock-L predicts novel binding pocket locations slightly more accurately than P2Rank for conventional docking with AutoDock-Vina. Paired with the observation that ablating structural cluster training (SCT) from DiffDock yields considerably degraded DockGen performance, these findings support the idea that SCT provides DL docking methods with useful knowledge for generalizing to novel binding pockets.

Unintuitively, DiffDock-L's results with protein-flexible relaxation applied post-prediction (i.e., DiffDock-L-Relax-Prot) demonstrate that fixed-protein relaxation (albeit unideal from a theoretical e.g., protein side chain perspective (Wankowicz et al., 2022)) yields less accuracy degradation to DiffDock-L's original ligand conformations compared to protein-flexible relaxation. Lastly, we note that none of the baseline methods could generate *any* PB-valid ligand conformations, suggesting that all of their "correct" poses are approximately accurate yet physically implausible in certain measurable ways.

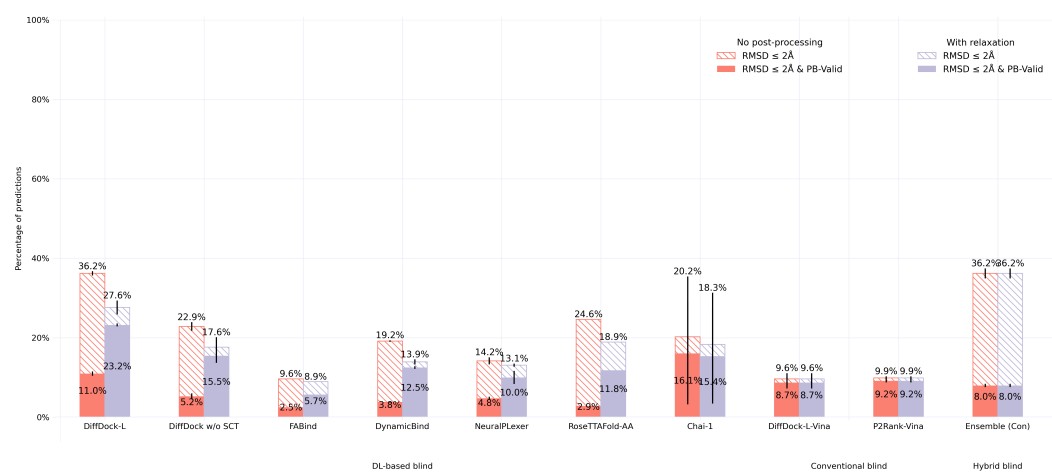

Figure 15: Pocket-only PoseBusters dataset results for successful single-ligand docking with relaxation.

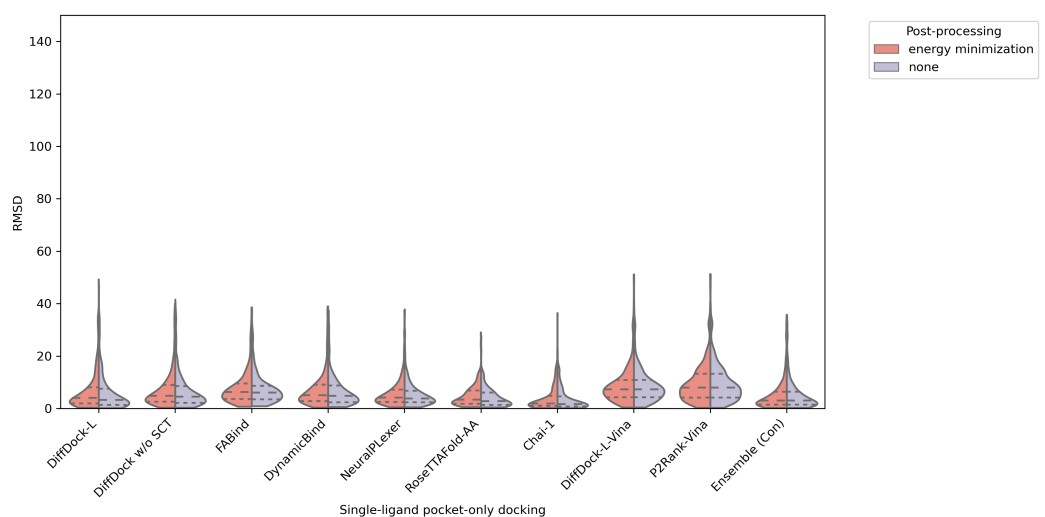

Figure 16: Pocket-only PoseBusters dataset results for single-ligand docking RMSD.

## H.2 EXPANDED ASTEX & POSEBUSTERS RESULTS

### H.2.1 POCKET-ONLY POSEBUSTERS RESULTS

Figures 15 and 16 illustrate the impact of reducing the binding pocket search space of each baseline docking method by providing each method with alternative versions of the predicted PoseBusters Benchmark protein structures that have been cropped to contain only ligand-interacting ($< 4$ Å heavy atom distance) protein residues and their (7) sequence-adjacent neighbors. Overall, we find that performing such *pocket-level* docking increases the docking success rates and favorably narrows the ligand RMSD distributions of DiffDock-L, DynamicBind, RoseTTAFold-AA, AutoDock Vina (w/ either DiffDock-L or P2Rank's predicted binding pockets), and Ensemble (Con), whereas for all other baselines, performance is either maintained or degraded marginally. This finding highlights that methods such as DiffDock-L and RoseTTAFold-AA are better at leveraging a reduced (e.g., structural) search space for each ligand conformation compared to other baseline methods such as Chai-1 and NeuralPLexer.

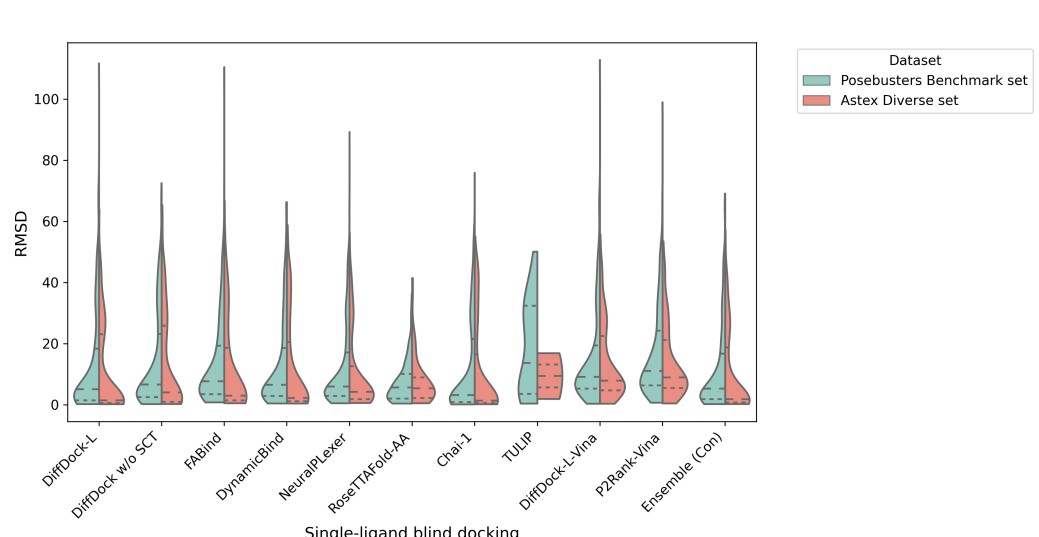

Figure 17: Astex & PoseBusters dataset results for single-ligand docking RMSD.

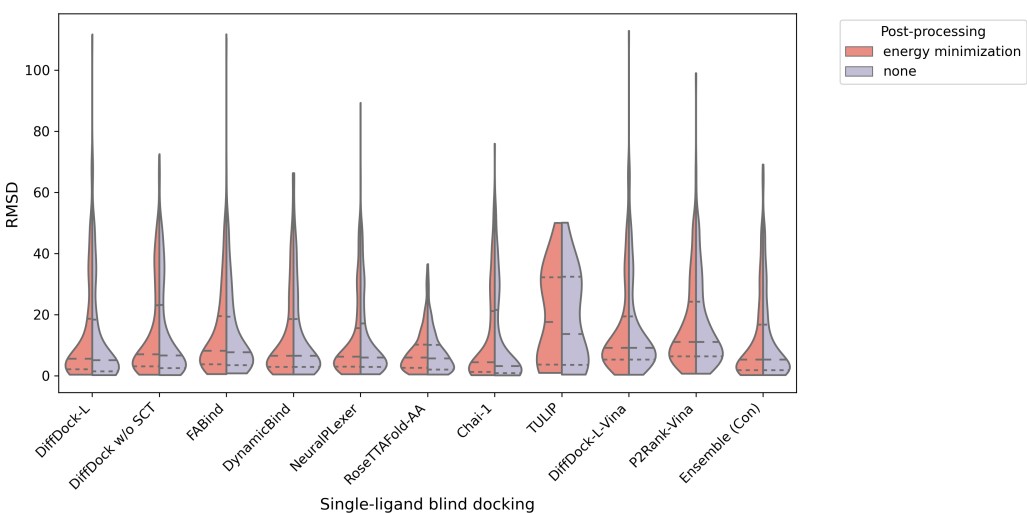

Figure 18: PoseBusters dataset results for single-ligand docking RMSD with relaxation.

### H.2.2 ASTEX & POSEBUSTERS RMSD RESULTS

In Figures 17 and 18, we report the ligand RMSD values of each baseline method across the Astex Diverse and PoseBusters Benchmark datasets, with relaxation being applied in the context of the PoseBusters Benchmark dataset. In short, we see that most methods are relatively similar in terms of their ligand RMSD distributions, with RoseTTAFold-All-Atom and Ensemble (Con), however, offering more condensed distributions overall. Interestingly, for Astex Diverse, TULIP also appears to produce a uniquely confined ligand RMSD distribution.

## H.3 EXPANDED CASP15 RESULTS

### H.3.1 OVERVIEW OF EXPANDED RESULTS

In this section, we begin by reporting additional CASP15 benchmarking results in terms of each baseline method's multi-ligand RMSD and lDDT-PLI distributions as violin plots. Subsequently, we report successful ligand docking success rates as well as RMSD and lDDT-PLI results specifically for the single-ligand CASP15 targets. Lastly, we report all the above single and multi-ligand results specifically using only the CASP15 targets for which the ground-truth (experimental) structures are publicly available, to support reproducible future benchmarking and follow-up works.

### H.3.2 MULTI-LIGAND RMSD AND LDDT-PLI

To start, Figures 19 and 20 report each method's multi-ligand RMSD and lDDT-PLI distributions with and without relaxation. We see that NeuralPLexer and Ensemble (Con) produce the most tightly bound and accurate RMSD and lDDT-PLI distributions overall.

### H.3.3 ALL SINGLE-LIGAND RESULTS

Next, Figures 21, 22, 23, and 24 display each method's single-ligand CASP15 docking success rates, PoseBusters validity rates, docking RMSD, and docking lDDT-PLI distributions, respectively. In summary, we can make a few respective observations from these plots. (1) DiffDock-L and NeuralPLexer are the only DL methods capable of successfully docking any single-ligand CASP15 complexes. (2) AutoDock Vina produces the most PB-valid single-ligand complexes overall, with TULIP shortly behind. (3) DiffDock-L and AutoDock Vina appear to achieve the most tightly bound and accurate RMSD distributions. (4) In contrast to (3), only DiffDock-L-Vina appears to achieve top results in terms of lDDT-PLI compared to the other baseline methods.

### H.3.4 SINGLE AND MULTI-LIGAND RESULTS FOR *public* TARGETS

Lastly, for completeness and reproducibility, Figures 25, 26, 27, and 28 present corresponding multi-ligand results for the public CASP15 targets, whereas Figures 29, 30, 31, and 32 report corresponding single-ligand results for the public CASP15 targets. Overall, we observe marginal differences between the full and public CASP15 target results for multi-ligand complexes, since once again NeuralPLexer achieves top results in this multi-ligand context. However, we notice more striking performance drops between the full and public *single*-ligand CASP15 target results, suggesting that some of the private single-ligand complexes are easier prediction targets than most of the publicly available single-ligand complexes. In short, we find that DiffDock-L-Vina performs the best in this setting.

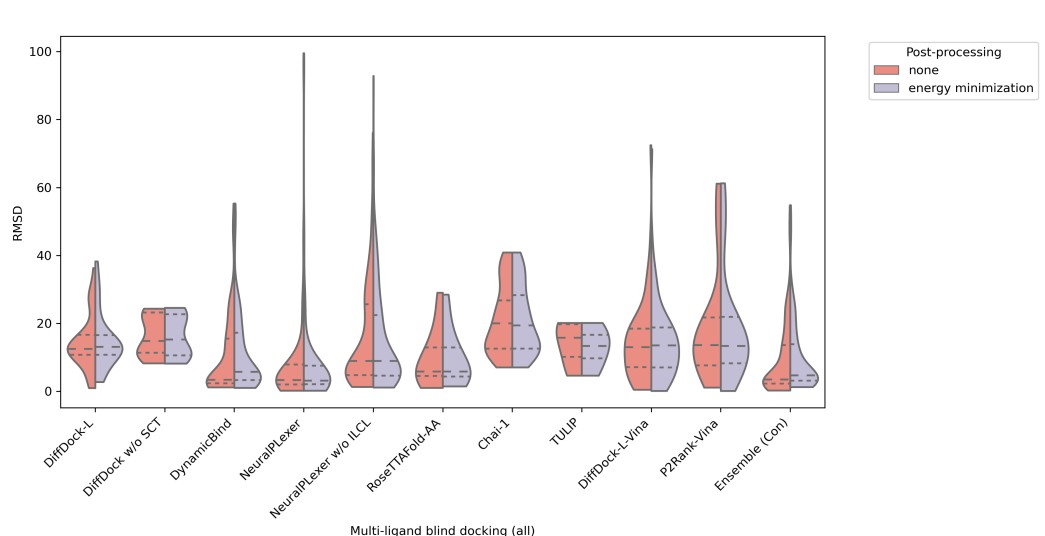

Figure 19: CASP15 dataset results for multi-ligand docking RMSD with relaxation.

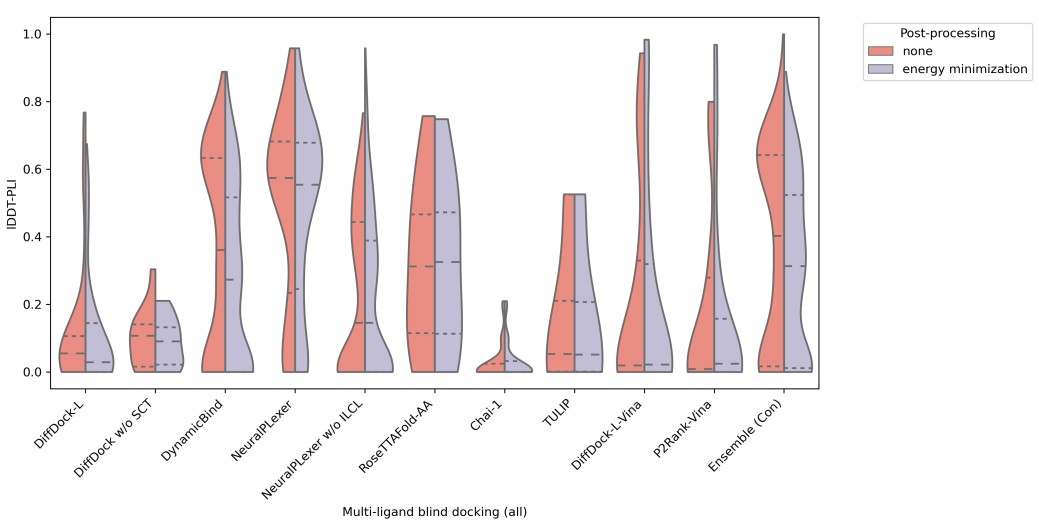

Figure 20: CASP15 dataset results for multi-ligand docking lDDT-PLI with relaxation.

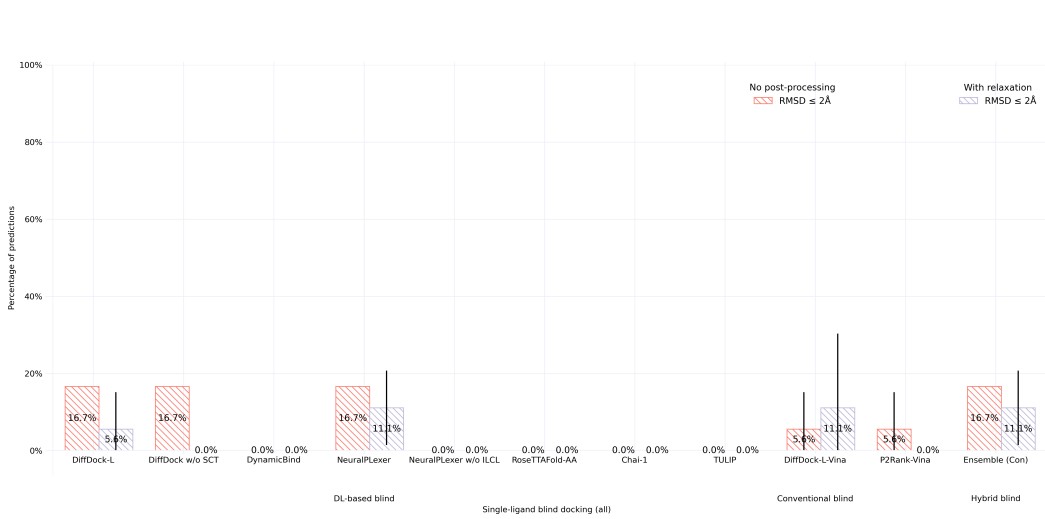

Figure 21: CASP15 dataset results for successful single-ligand docking with relaxation.

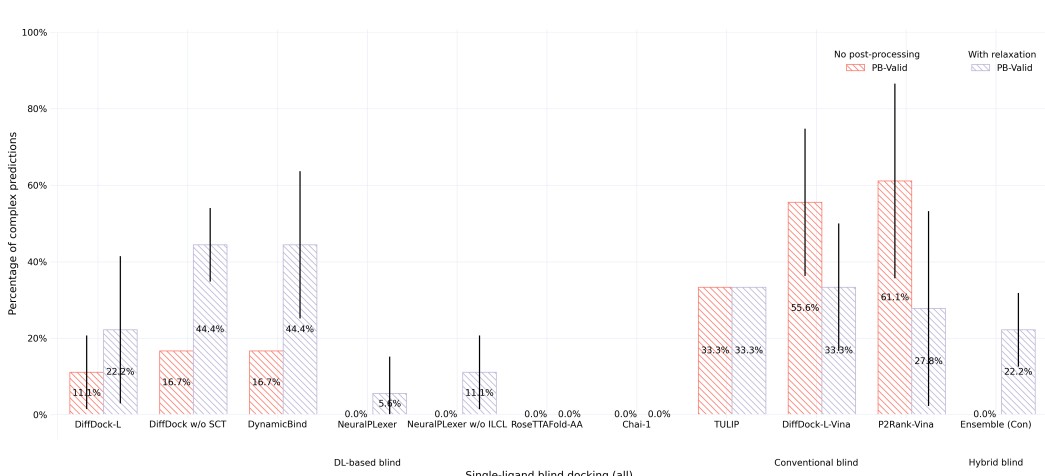

Figure 22: CASP15 dataset results for single-ligand PoseBusters validity rates with relaxation.

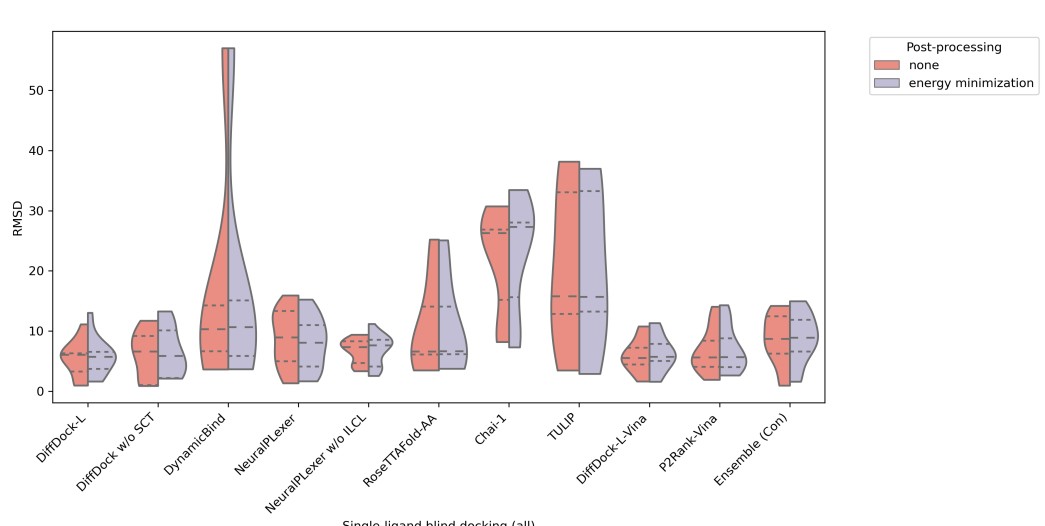

Figure 23: CASP15 dataset results for single-ligand docking RMSD with relaxation.

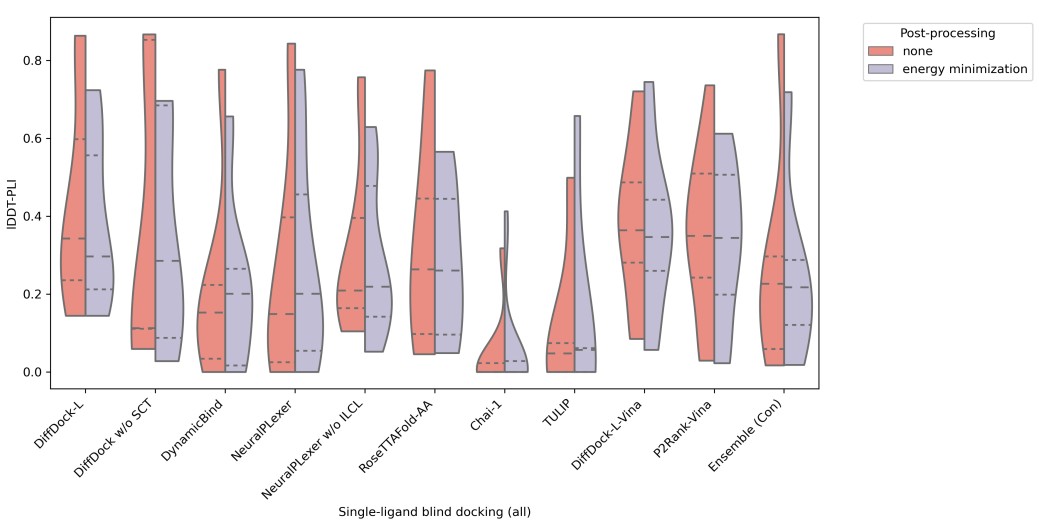

Figure 24: CASP15 dataset results for single-ligand docking lDDT-PLI with relaxation.

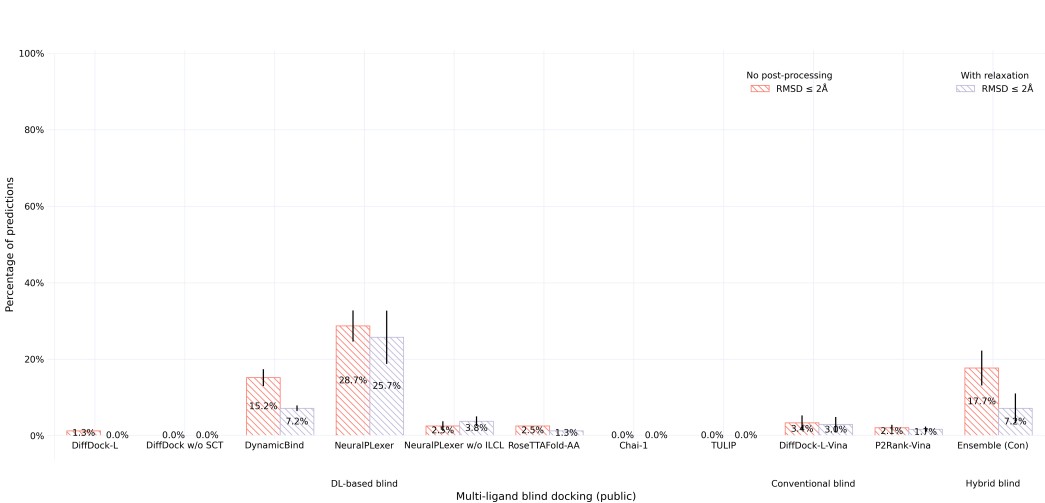

Figure 25: CASP15 public dataset results for successful multi-ligand docking with relaxation.

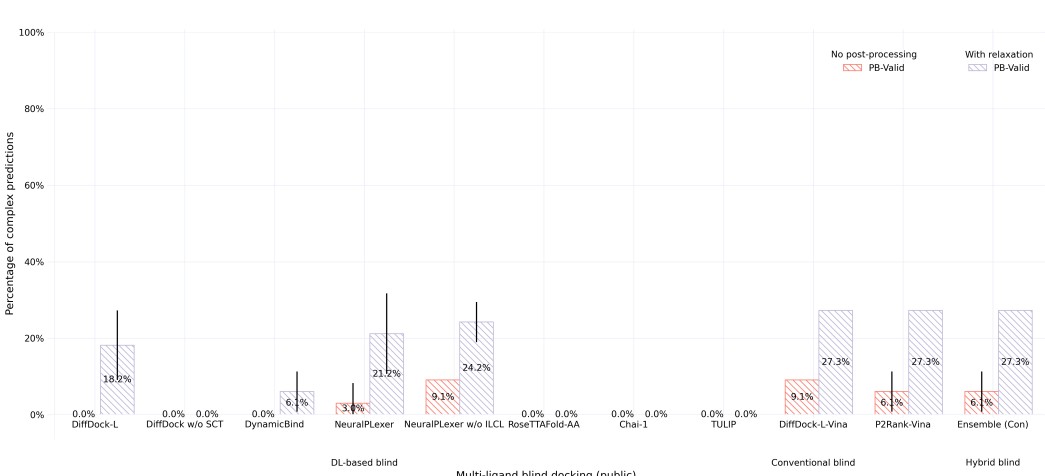

Figure 26: CASP15 public dataset results for multi-ligand PoseBusters validity rates with relaxation.

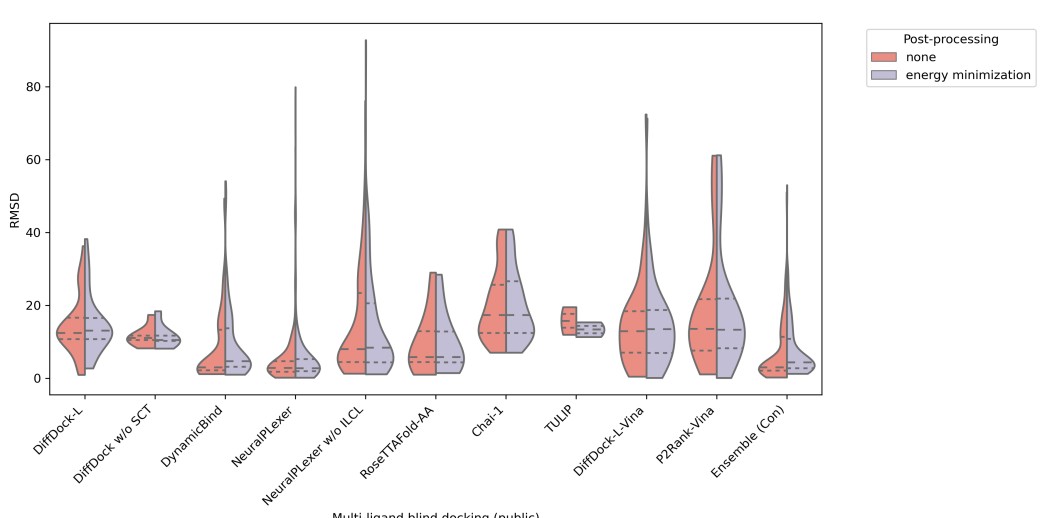

Figure 27: CASP15 public dataset results for multi-ligand docking RMSD with relaxation.

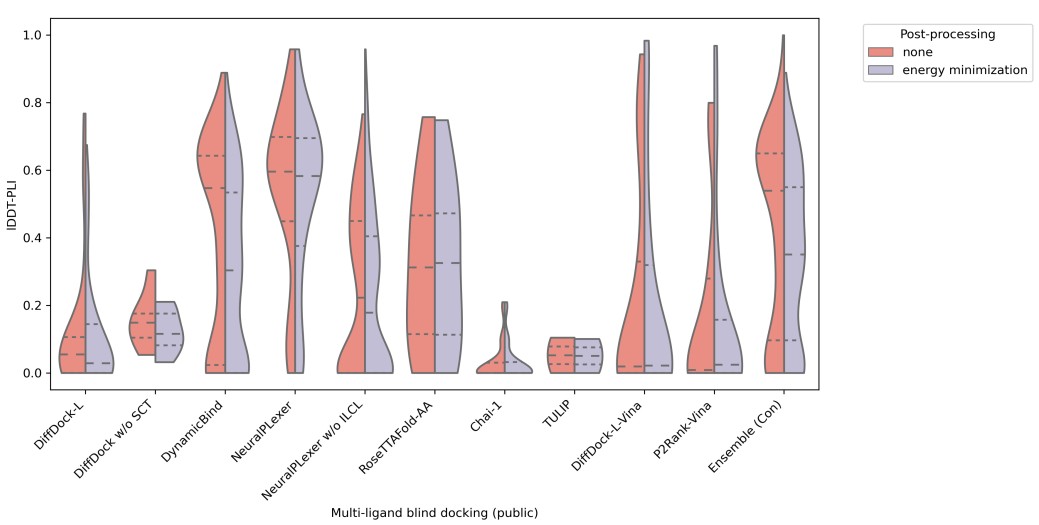

Figure 28: CASP15 public dataset results for multi-ligand docking lDDT-PLI with relaxation.

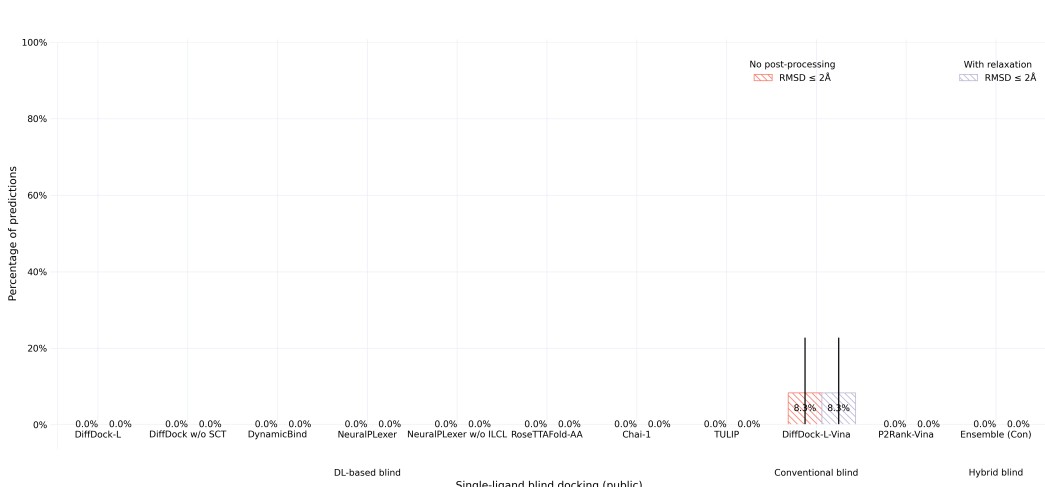

Figure 29: CASP15 public dataset results for successful single-ligand docking with relaxation.

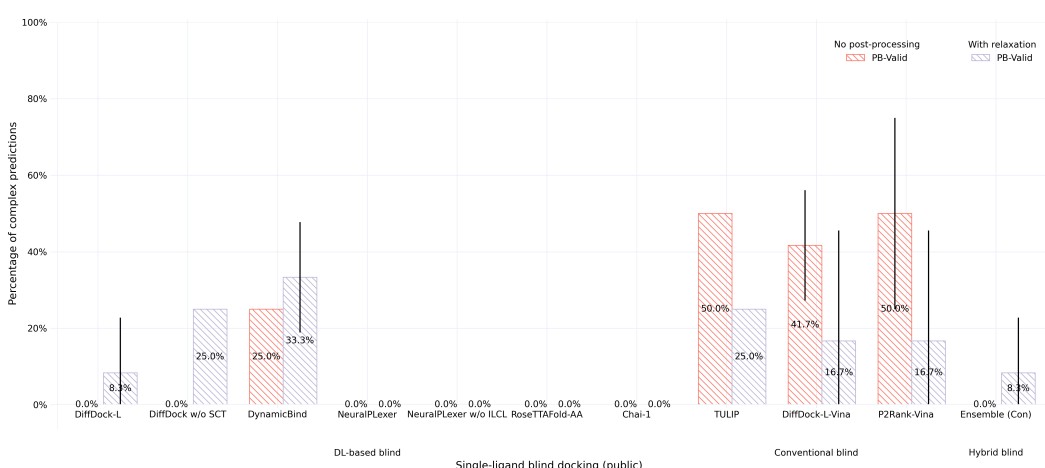

Figure 30: CASP15 public dataset results for single-ligand PoseBusters validity rates with relaxation.

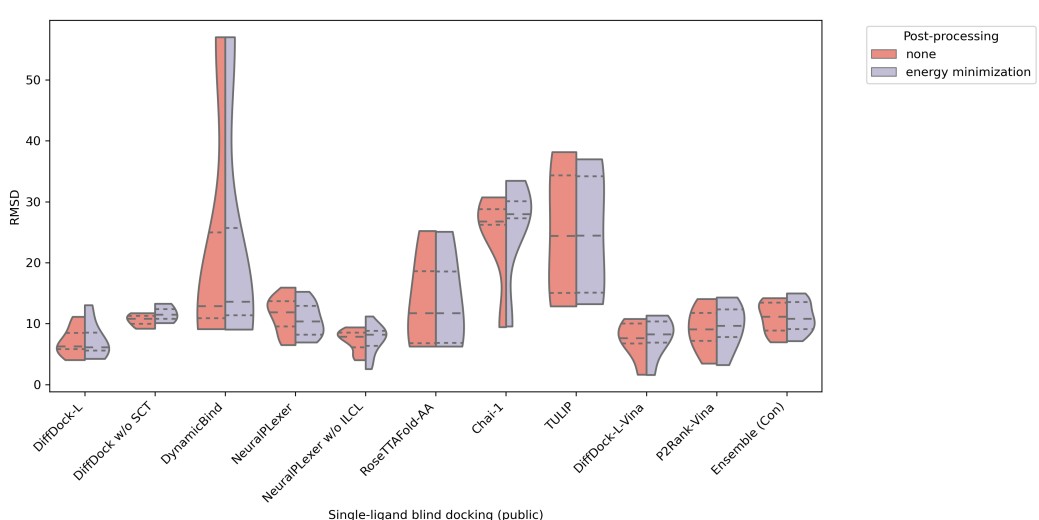

Figure 31: CASP15 public dataset results for single-ligand docking RMSD with relaxation.

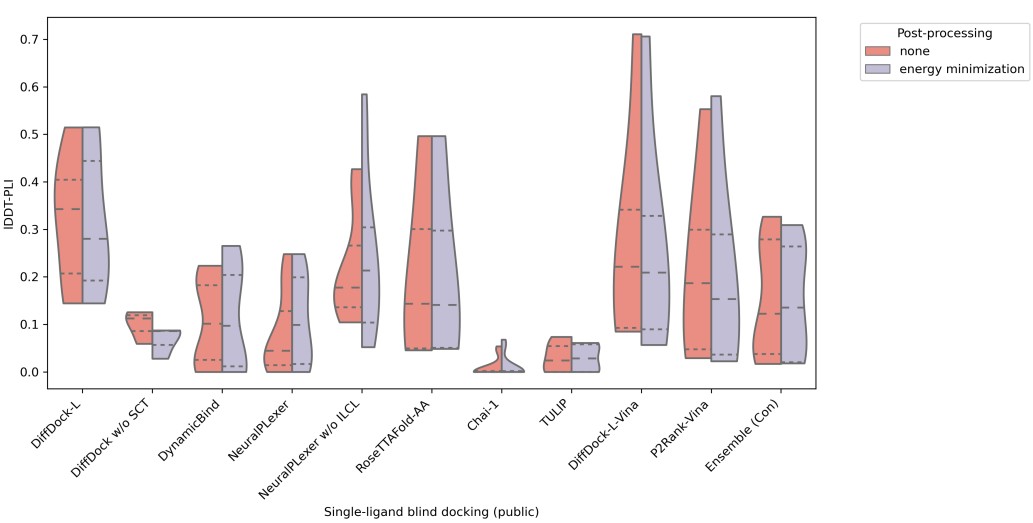

Figure 32: CASP15 public dataset results for single-ligand docking lDDT-PLI with relaxation.

