# OpenReview forum: "Deep Learning for Protein-Ligand Docking: Are We There Yet?"
_ICLR.cc/2025/Conference — ICLR 2025 Conference Withdrawn Submission_

### Official Review · Reviewer_BYat · 2024-10-28

**Soundness:** 2
**Presentation:** 2
**Contribution:** 2
**Rating:** 3
**Confidence:** 5

**Summary:**

The paper introduces POSEBENCH, a new benchmark suite designed to evaluate protein-ligand docking models, with extra focus on apo protein structures, covering both single- and multi-ligand docking scenarios, blind docking. Empirical findings suggest that while DL models outperform traditional docking techniques, their performance in multi-ligand docking is still limited. POSEBENCH aims to improve model generalization by using diverse, physics-informed datasets and provides an open-source tool to foster advancements in protein-ligand docking.

**Strengths:**

- Such evaluation is valuable.
- Code, data, and tutorials are provided openly, supporting reproducibility and further research.

**Weaknesses:**

- There aren’t many new conclusions for the community.
- Somewhat outdated: The PDBBind time-split set has been shown by works like AlphaFold 3 to be insufficient for docking tasks. The authors’ comparisons with this training set limit the evaluation's scope.
- The datasets used are essentially from the RCSB PDB. It would be more meaningful if the authors curated a new dataset following their own pipeline; analyzing pre-existing data is not a truly systematic evaluation.
- The multi-ligand dataset is too small (only 19 complexes), and similarly, they are also sourced from RCSB PDB. The PoseBusters PDB also has many multi-ligand cases from RCSB PDB that could be included.
- Results are not well-presented:
  - The authors introduce four datasets as parallel, but, for example, DockGen results are absent from the main text, while others take up too much space with excessive whitespace.
  - Input and output format descriptions are unnecessary in the main text.
- Section 3.2 is incorrect; centroid distance is the metric for pocket identification, and site-specific docking measures docking within the correct pocket.

**Questions:**

- Why is the Astex Diverse dataset referred to as an upper bound? Is there additional data/theoretical support?
- How is it ensured that the AF3-predicted structure is used as an apo structure?
- Why does Chai-1 perform so poorly on PoseBusters, despite its claims of outperforming AF3?

---

### Official Review · Reviewer_QTvg · 2024-10-29

**Soundness:** 2
**Presentation:** 3
**Contribution:** 1
**Rating:** 3
**Confidence:** 3

**Summary:**

For the current challenges of DL application in the field of biological applications, in particular the modeling of protein-ligand docking, the authors introduced the unified benchmark for DL-based methods evaluation. PoseBench consists of 4 datasets - Astex Diverse dataset, PoseBusters Benchmark dataset, DockGen and CASP15 dataset. Performance of several models was evaluated on single-ligand and multi-ligand docking tasks. The main conclusion obtained is that most DL models struggle to generalise to multi-ligand docking targets.

**Strengths:**

- Conducted evaluation experiments of the existing DL docking methods for protein-ligand docking and protein-ligand structure generation tasks.
 - Described the limitations of those methods and their inability to generalize well to multi-ligand protein targets.
 - Proposed the direction for future works and studies.
 - Proposed generalized evaluation metrics.

**Weaknesses:**

Seems that the major contribution in the proposed benchmarking dataset was already produced by works cited in paper (e.g. sources listed in Table 1) or augmented with predictions from AlphaFold3. The only visible contribution is filtering out complexes and alignment of predictions from AlphaFold3 with the ground-truth protein-ligand structures.

**Questions:**

The main question I'd like to address is the usage of AlphaFold3 for prediction of complex structure for each protein. For applications such as drug design it seems that augmentation of datasets using generated data can lead to accumulation of not-necessarily-physical results being treated as the "ground truth", and therefore, future DL methodologies would fail to predict real stable structures. How would you justify the usage of AlphaFold3 for data augmentation?

---

### Official Review · Reviewer_K33a · 2024-10-31

**Soundness:** 2
**Presentation:** 2
**Contribution:** 2
**Rating:** 5
**Confidence:** 2

**Summary:**

The authors propose a benchmark for broadly applicable protein-ligand docking. Studying protein-ligand binding effect is crucial in drug discovery and in the last few years several deep learning based methods have been developed to predict protein structures using sequence, MSA etc. So it becomes even more important to study docking using predicted protein structures, and also when the binding pockets are unknown. The authors have identified several state of the art methods and datasets to compare deep learning based models to conventional docking models. The observation that the recent deep learning methods combined with structural relaxation outperforms conventional single ligand docking and multi-modal training on protein sequence and structure has promise corresponds to the research on multimodal protein model studies. The authors recommend using physics-informed inter-ligand steric clash penalties to improve deep learning docking methods for multi-ligand docking targets. The paper is well written and is a useful survey to get initial research directions.

**Strengths:**

The selection of both deep learning based models and conventional models provide a fair comparison in answering the key question authors ask. Detailed anlaysis of each method is done to understand the advantages and disadvantages. Key observations are converted to actions/recommendations for interested researchers.

**Weaknesses:**

Figure 2 and Figure 3 are hard to read and could be improved with larger fonts for xlabel, ylabel and legends.

The recommendation to train new docking methods directly on structurally clustered multi-ligand protein complexes is not novel and recent multi-modal protein models have already started training on protein structures. For example in the ESM3 paper (https://www.biorxiv.org/content/10.1101/2024.07.01.600583v1) the authors introduce multiple tracks (sequence, structure, SASA, annotations, SSR) as multiple modalities and trained on protein model together. They also train on multiple chains and introduce a chain breaker as a token. It might be useful to explore protein surface (specifically properties extracted from side chain atoms and not just core atoms) as part of training docking methods and compare the sequence-structure to sequence-structure-surface model performances.

**Questions:**

It was interesting to see Chai-1 (along with DiffDock-L) achieve the best overall performance across both of single-ligand datasets in percentage of correct and valid generated ligand poses. Do the authors have an insight on the what features of the model helped to achieve the performance?
For example does Chai-1 has the advantage that it can use MSAs, templates, multi-chain protein sequence and fragmented ligand smiles strings as compared to the other methods?

---

### Official Review · Reviewer_2qyZ · 2024-11-01

**Soundness:** 3
**Presentation:** 3
**Contribution:** 2
**Rating:** 5
**Confidence:** 3

**Summary:**

This paper proposes a benchmark (combining existing data) for protein-ligand docking on deep learning methods. There are three settings: using predicted protein structures, multiple ligand docking and blind docking. The benchmark evaluates both recent deep learning methods and conventional approaches for single and multi-ligand protein docking.

**Strengths:**

1. The paper focuses on several factors for practical drug discovery (using predicted structure, no known binding pocket, multiple ligands).
2. The results include extensive evaluation with SOTA methods.

**Weaknesses:**

1. Why blind docking? The cases where the binding site is known are more practical.
2. Chai-1 is basically AF3. Did you use MSA?
3. Using AlphaFold3 structure as apo structures is questionable. Why not use d3pm or APObind?
4. The number of multi-ligand docking targets is small and they are all from the CASP15 dataset. As CASP15 targets typically have low homology and low-quality MSAs, it confounds the conclusion for multi-ligand docking. Is the poor performance due to multi-ligand or difficult protein targets?

**Questions:**

See weakness

---

### Official Review · Reviewer_8dEe · 2024-11-02

**Soundness:** 1
**Presentation:** 2
**Contribution:** 1
**Rating:** 3
**Confidence:** 4

**Summary:**

This manuscript presents a benchmark on protein-ligand docking. They are curated from four datasets: Astex Diverse, PoseBusters, DockGen and CASP15. The authors have also demonstrated the applicability of three types of methods in the benchmark: traditional methods, predictive methods and generative methods.

**Strengths:**

The experiments are done with different docking methods to show their effectiveness on the benchmarks.

The authors have also showed the effectiveness of diverse cluster training and use of physics informed clash penalization.

**Weaknesses:**

All the docking datasets that are used to create the benchmark by merging them together are previously proposed, all validated by researchers curating them from literature or via competitions. Thus the core contribution to the introduction of the benchmark remains unclear to the reviewer.

The protein structures are determined by Alpha Fold without any rationales being given to support this selection.

Several structures or proteins were eliminated from each of the datasets due to limitations on the computing. Mention on the distribution of the length of these and the remaining structures were needed. Are these structures available? Or only criteria is that AlphaFold on the given machine could not perform prediction due to limitation is not clear to the reviewer.

**Questions:**

Please answer to the weakness part.

In addition, in the results CASP15 seems to provide outliers for most of the experiments. Is there any explanations for this?

---

### Official Review · Reviewer_wEEA · 2024-11-03

**Soundness:** 2
**Presentation:** 2
**Contribution:** 3
**Rating:** 5
**Confidence:** 4

**Summary:**

The manuscript introduces PoseBench, a novel and comprehensive benchmark designed to assess deep learning (DL) methods for protein-ligand docking. The primary focus of PoseBench is to address the limitations of existing benchmarks by evaluating docking methods under conditions that more closely mimic real-world applications. These conditions include the use of predicted (apo) protein structures, concurrent docking of multiple ligands to a single target protein, and scenarios where binding pockets are unknown. While the benchmark is a simple collection of 4 existing datasets, the availability of code and dataset is a plus.

**Strengths:**

1. The authors have compiled a benchmark by collecting 4 datasets, Astex Diverse, PoseBusters Benchmark, DockGen, and CASP15. They have performed comprehensive tests of DL, conventional, and hybrid methods.
2. They have indicated that DL methods outperform conventional methods, and that multi-ligand is a blind point for DL methods.
3. They provide comprehensive data and codes.

**Weaknesses:**

However,
1. this paper doesn't bring too much new things: the conclusions are somewhat controversial or well-known. For example, the comparison of DL and docking methods strongly depends on the settings, and many studies obtained different or contrary conclusions. This  is one of many papers supporting DL methods.
2. The paper claims "no prior works have systematically studied the behavior of docking methods".  Obviously this is over-claimed. All published papers have tried to answer this question, e.g PoseBusters. Though the authors have doubled the dataset, this is incremental change as a benchmark test. The authors didn't indicate anything new from the expanded dataset.
3. The thinking on the physics-informed loss function is interesting, but it was from a simple ablation. This is definitely not enough since NeuralPLexer was tuned for the whole settings.

**Questions:**

The authors may consider adding the physics-informed loss function into other packages.

---

### Official Review · Reviewer_rWM5 · 2024-11-04

**Soundness:** 3
**Presentation:** 2
**Contribution:** 3
**Rating:** 5
**Confidence:** 3

**Summary:**

This paper introduces a benchmark for protein-ligand docking and structural generation, focusing on three aspects: (1) docking using predicted protein structures, (2) docking multiple ligands concurrently, and (3) docking without a known binding pocket. The authors conducted experiments on several models, presented their findings, and discussed potential future directions for this area.

**Strengths:**

The aspects the authors focus on are of practical significance. The article is well-structured and easy to follow.

**Weaknesses:**

1. The authors did not first address the limitations of previous benchmarks or explain how their new benchmark resolves these issues. While the facets introduced are practically relevant, their suitability for inclusion in a docking benchmark could be questioned. Using predicted protein structures, docking multiple ligands simultaneously, and docking without a known binding pocket are not universal needs in docking tasks. It may be overly broad to expect all docking models to address each of these facets.
2. The title, while eye-catching, may create some inconsistency with the findings of the paper: the authors gave a negative answer in line 500 to the question posed by the title, "Deep Learning for Protein-Ligand Docking: Are We There Yet?", despite stating in line 026 that "DL methods consistently outperform conventional docking algorithms." As the paper primarily introduces new aspects to the general docking task, the title may not fully reflect this focus.
3. The presentation of the paper could be improved: (a) Figures 2-5 could be more polished, and the text within these figures is small relative to the main text, affecting readability; (b) there is a typographical error in "PDBbind".

**Questions:**

1. The effect of apo versus holo structures on docking is indeed an important area to emphasize especially in the search for effective models for drug development. However, it’s unclear if the authors’ definition of "apo" aligns with the field. Does "apo" in this context refer simply to the absence of a ligand in the pocket area? Typically, the primary difference involves the side-chain conformation within the pocket, which may change with ligand binding. Predicted protein structures could reflect a range of conformations, including holo or intermediary states, rather than strictly apo or holo.
2. Does the use of AlphaFold’s predictions for docking comply with its terms of use? (See: https://alphafoldserver.com/about)

---

### Official Review · Reviewer_4wns · 2024-11-11

**Soundness:** 2
**Presentation:** 3
**Contribution:** 2
**Rating:** 6
**Confidence:** 4

**Summary:**

The paper introduces a new protein ligand benchmark:

* Data: It uses existing established datasets (posebusters, casp15 etc) and also adds apo data to them by using AF3, compares different datasets in terms of pocket-ligand interaction similarity.

* Benchmarking tasks: It targets well established benchmarking task of single ligand docking and adds a multi ligand task, including molecular fragment docking.

* Benchmarked models: A wide and diverse set of methods across different categories, from more traditional to more recent generative ones:  ADVina, Tulip, FABind, RF-AA, DynamicBind, NeuralPLexer, Chai-1, DiffDock-L.

* Benchmark evaluations: The well-accepted RMSD <2A criterion and PoseBusters structural validity is used for single ligand evals, for multiligand docking CASP15 criteria (% RMSD<2A and lDDT-PLI) are given.

* Results: Single ligand docking case - 1) training on diverse data boosts generalizability. 2)Generative methods outperform conventional docking tools in single ligand docking. 3)in multi ligand or fragment docking case, the situation is more complex, success rate of all models are lower, NeuralPlexer seems to do better than others, authors attribute this to the steric clash loss used in this model.

**Strengths:**

The following are the contributions that make this paper significant to the field of protein-ligand docking
* Predicted protein unbound (predicted-apo) state benchmark
* Multiligand/fragment docking benchmark
* Use of relatively recent models like DiffDock-L and Chai-1 in the benchmarks as well as AF3 structures
* Hybrid models such as combination of different pocket-finder ML tools with pocket-requiring conventional tools.


This is because benchmarks of these kinds have mostly been performed within model development efforts and not systematically across models.

**Weaknesses:**

* There was been a recent attempt at extensive docking benchmarking called _Plinder_ and it brought an important understanding to the field about the amount of leak from train to widely used benchmark sets. There is no analysis in this work that makes use of that highly relevant work, one would expect at least a leak measurement for different ML models.
* Despite apo state benchmark being the strength of this paper, there is no apo vs holo analysis in the main text. Some relevant context is given in the appendix. Should be summarized in the main text, e.g. analysis based on apo-to-holo RMSD of the _pocket_
* In multiple occasions important nuance about results is not given in text, I will highlight few examples:
  * Chai-1 significantly outperforms others in the single ligand docking task if PB validity is to be considered as a metric, however the same model performs very poorly in the DockGen dataset according to Fig 13 in Appendix; and in pocket-only case, results get better however the error bar is pretty large. So although the final conclusion is correct, that “are we there yet?” question is answered negatively, details like this should be surfaced from appendix.
  * Authors claim both DiffDock-L and Chai-1 perform best in single-ligand docking but Chai-1 should be singled out looking at the results shown in the main text. It is only in appendix with DockGen results that DiffDock-L appears to perform better than others. This should be made more explicit in the main text.
  * Multi-ligand case: NeuralPlexer’s steric clash loss is claimed to be responsible for its success but it is not mentioned that pose validity increases without this loss (Figs 4&5). Furthermore, it is again left to appendix (Fig 12) that the reader can see that NeuralPlexer does not reproduce the protein-ligand interaction profile significantly better than other models, even though authors highlight the mild similarity in hydrophobic interaction profile, which reads more like a correlation than causation. Because if there was causation DiffDock w/o SCT should have been the second best model.
  * Relaxation: It is not mentioned that how the relaxation is performed can have a significant impact on results: Ideally, if relaxation could be carried out accurately to find the global minimum within a pocket, all models that can locate the pocket would have yielded the same global minimum pose. The rigidity of the molecule potential, how the relaxation algorithm handles cases with clashes etc. determines how much the initial pose estimate of the model matters for the final relaxed pose. Nevertheless, readers would appreciate the relaxation results, as they provide a practical picture of what is used in the field today but should not be over-interpreted.

**Questions:**

Some questions picked from limitations section:
* For models that can perform blind docking, what is the success rate in finding the pocket at all?
* What's the leak between train set of these models with the test sets mentioned here?
* Can you report a ligand/fragment type-resolved analysis of multiligand case? It would be helpful to see whether fragments or ions etc is contributing to the low success rate more than others.

---

### Note · Authors · 2024-11-25

**Comment:**

We authors sincerely thank each reviewer for their detailed and insightful feedback on our paper. Based on the reviewers' comments, and given the rapid pace of development in this field (e.g., with the recent release of AlphaFold 3's local inference code), we believe the community would be better served by an updated and expanded version of this benchmark (informed by the reviewers' feedback) that will take more time to complete than allotted for rebuttals. As such, we will unfortunately be withdrawing the paper at this time. Nonetheless, we would encourage the reviewers to keep an eye out for our updated benchmark in the coming month and to engage with our publicly available resources to help us improve them iteratively over time.

**Withdrawal Confirmation:**

I have read and agree with the venue's withdrawal policy on behalf of myself and my co-authors.